



# A multimodel evaluation of the potential impact of shipping on particle species in the Mediterranean Sea

Lea Fink[1], Matthias Karl[1], Volker Matthias[1], Sonia Oppo[2], Richard Kranenburg[3], Jeroen Kuenen[3], Sara Jutterström[4], Jana Moldanova[4], Elisa Majamäki[5], Jukka-Pekka Jalkanen[5]

[1]Helmholtz-Zentrum Hereon, Institute of Coastal Environmental Chemistry, 21502 Geesthacht, Germany
[2]AtmoSud, Air Quality Observatory in the Provence-Alpes-Côte d'Azur region, 13006 Marseille, France
[3]TNO, Netherlands Organization for Applied Scientific Research, 3584 CB Utrecht, The Netherlands
[4]IVL, Swedish Environmental Research Institute, 411 33 Göteborg, Sweden
[5]FMI, Finnish Meteorological Institute, FI-00560 Helsinki, Finland

*Correspondence to*: Lea Fink (lea.fink@hereon.de)

**Abstract.** Shipping contributes significantly to air pollutant emissions and atmospheric particulate matter (PM) concentrations. At the same time worldwide maritime transport volumes are expected to continue to rise in the future. The Mediterranean Sea is a major short-sea shipping route within Europe, as well as the main shipping route between Europe and East Asia. As a result, it is a heavily trafficked shipping area, and air quality monitoring stations in numerous cities along the Mediterranean coast have detected high levels of air pollutants originating from shipping emissions.

The current study is a part of the EU Horizon 2020 project SCIPPER (Shipping contribution to Inland Pollution - Push for the Enforcement of Regulations) which intends to investigate how existing restrictions on shipping-related emissions to the atmosphere ensure compliance with legislation. To demonstrate the impact of ships on relatively large scales, the potential shipping impacts on various air pollutants can be simulated with chemistry transport models.

To determine formation, transport, chemical transformation and fate of $PM_{2.5}$ in the Mediterranean Sea in 2015, five different regional chemistry transport models (CAMx – Comprehensive Air Quality Model with Extensions, CHIMERE, CMAQ – Community Multiscale Air Quality model, EMEP – European Monitoring and Evaluation Programme model, LOTOS-EUROS) were applied. Furthermore, $PM_{2.5}$ precursors ($NH_3$, $SO_2$, $HNO_3$) and inorganic particle species ($SO_4^{2-}$, $NH_4^+$, $NO_3^-$) were studied, as they are important for explaining differences among the models. STEAM version 3.3.0 was used to compute shipping emissions, and the CAMS-REG v2.2.1 dataset was used to calculate land-based emissions for an area encompassing the Mediterranean Sea at a resolution of $12 \times 12$ km$^2$ (or $0.1° \times 0.1°$). For additional input, like meteorological fields and boundary conditions, all models utilized their regular configuration. The zero-out approach was used to quantify the potential impact of ship emissions on $PM_{2.5}$ concentrations. The model results were compared to observed background data from monitoring sites.

Four of the five models underestimated the actual measured $PM_{2.5}$ concentrations. These underestimations are linked to model-specific mechanisms or underpredictions of particle precursors. The potential impact of ships on the $PM_{2.5}$ concentration is



between 15 % and 25 % at the main shipping routes. Regarding particle species, $SO_4^{2-}$ is main contributor to the absolute ship-related $PM_{2.5}$ and also to total $PM_{2.5}$ concentrations. In the ship-related $PM_{2.5}$, a higher share of inorganic particle species can be found when compared to the total $PM_{2.5}$. The seasonal variabilities in particle species show that $NO_3^-$ is higher in winter and spring, while the $NH_4^+$ concentrations displayed no clear seasonal pattern in any models. In most cases with high concentrations of both $NH_4^+$ and $NO_3^-$, lower $SO_4^{2-}$ concentrations are simulated. Differences among the simulated particle species distributions might be traced back to the aerosol size distribution and how models distribute among the coarse and fine mode ($PM_{2.5}$ and $PM_{10}$). The seasonality of wet deposition follows the seasonality of the precipitation, displaying that precipitation predominates the wet deposition.

### List of Abbreviations

| | |
|---|---|
| 1.5-D | 1.5-dimensional |
| CAMx | Comprehensive Air Quality Model with Extensions |
| CMAQ | Community Multiscale Air Quality model |
| CTM | Chemistry Transport Model |
| EEA | European Environment Agency's |
| EMEP | European Monitoring and Evaluation Programme model |
| GNFR | Gridded Nomenclature for Reporting |
| MARS | Model for an Aerosol Reacting System |
| MEPC | Marine Environment Protection Committee |
| NMB | Normalized mean bias |
| NMVOC | Non-methane volatile organic compound |
| PM | particulate matter |
| POA | primary organic aerosol |
| PSAT | Particulate Source Apportionment Technology |
| R | Spearman's correlation coefficient |
| RADM-AQ | Regional Acid Deposition Model−Aqueous Chemistry |
| RMSE | root mean square error |
| SCIPPER | Shipping contribution to Inland Pollution - Push for the Enforcement of Regulations |
| SOAP | Secondary Organic Aerosol Processor |
| SOAs | Secondary organic aerosol |
| STEAM | Ship Traffic Emission Assessment Model |
| CBM-IV | Carbon bond mechanism (version IV) |
| TPPM | Total primary particulate matter |



| VBS | volatility basis set |
| --- | --- |
| VOC | volatile organic compound |

## 1 Introduction

Exhaust particles emitted from shipping have a large share in total emissions from the transport sector (Corbett and Fischbeck,
1997; Eyring et al., 2005), thereby affecting the chemical composition of the atmosphere as well as the regional air quality.
Particularly in coastal areas, maritime transport contributes a considerable fraction to air pollution (Viana et al., 2014).
High $PM_{2.5}$ concentrations can be caused by transported particles, desert dust or the production of secondary particulate matter
(Tomasi et al., 2017). Previous studies have revealed that in Europe, the $PM_{2.5}$ (particulate matter < 2.5 µm) concentration
increase caused by shipping emissions is small (Viana et al., 2009; Aksoyoglu et al., 2016). Nevertheless, in the Mediterranean
region the relative ship impact on the $PM_{2.5}$ concentration is large, with a share of 5 % to 20 % of the total $PM_{2.5}$ concentration
(e.g., Aksoyoglu et al. 2016, Nunes et al., 2020). Especially the formation of secondary particulate matter from ship emissions
is of importance. According to Viana et al. (2009), the secondary contribution of ship emissions is equivalent to double their
primary contribution. Secondary particles in the atmosphere form from gaseous precursors, whereas primary particles are
directly emitted and evolve within a short time to form secondary particles. To improve the air quality in coastal regions, it is
important to identify the pollutant sources and make reliable estimations of their impacts on surrounding PM levels. It has
been shown that the majority of secondary particles contributing to local PM in ports come from shipping (Song & Shon,
2014). Furthermore, according to Klimont et al. (2017), the proportion of international shipping's particulate matter primary
emissions to global anthropogenic emissions is between 3% and 4%, which is comparable to road traffic. Additionally,
shipping contributions to total $PM_{2.5}$ concentrations far from coastlines were found to be responsible for exceedances of WHO
air quality guideline values (Nunes et al., 2020). The annual mean $PM_{2.5}$ limit value in the EU is 25 µg/m³ (EU DIRECTIVE
2008/50/EC, 2008), whereas the annual mean $PM_{2.5}$ goal established by the WHO is 5.0 µg/m³ (WHO, 2021). Strong evidence
was found between exposure to $PM_{2.5}$ and the occurrences of certain diseases affecting the lungs, cancer, or type 2 diabetes
(Heusinkveld et al., 2016; Chen et al., 2016; Gao and Sang, 2020). According to the WHO, there is no safe level of $PM_{2.5}$;
thus, the gap between the WHO and EU $PM_{2.5}$ values is of actual concern (Karamfilova, 2022).
The MEPC decided in December 2022 to establish a sulfur emission control area in the Mediterranean Sea by 1st January 2025.
In this area, the limit for sulfur in fuel oils used on board ships is 0.10 % (IMO, 2022). The global sulfur cap for marine vessels
came into effect in January 2020, which declares that the sulfur content of any fuel oil used from ships must not exceed 0.50
% m/m, except for ships using 'equivalent' compliance mechanisms, such as scrubbers. Calculations show that this policy has
led to $PM_{2.5}$ reductions ranging from 0.5 µg/m³ to more than 2.0 µg/m³ along the major shipping routes in the Mediterranean
Sea (Jonson et al., 2020). These relatively strict 2020 regulations are expected to lower the number of $PM_{2.5}$-related premature
deaths by on average 15% (Viana et al., 2020).



Although the Mediterranean Sea contains one of the busiest shipping routes worldwide only a few regional-scale chemistry transport modeling studies have considered this region. Viana et al. (2014) reviewed studies concerning the impacts of shipping emissions on air quality in European coastal areas, noting that the highest $PM_{2.5}$ contributions were found in the Mediterranean Sea and North Sea. Aksoyoglu et al. (2016) studied $PM_{2.5}$ concentrations in the Mediterranean Sea followed up by a comparison of two models. Marmer and Langmann (2005) investigated the Mediterranean Sea on a broader scale and without comparing different CTM systems. Nevertheless, other studies have concentrated on smaller domains, such as the Iberian Peninsula (Baldasano et al., 2011; Nunes et al., 2020), the eastern Mediterranean Sea with the Arabian Peninsula (Večeřa et al., 2008; Tadic et al., 2020; Celik et al. 2020; Friedrich et al., 2021), or urban scale and harbor cities (Schembari et al., 2012; Donateo et al., 2014; Prati et al., 2015). None of these studies, however, analyzed the potential shipping impacts on $PM_{2.5}$ concentrations together with individual aerosol species on a regional basis, additionally comparing the results of five CTMs.

A wide range of gaseous pollutants, such as sulfur dioxide ($SO_2$) and nitrogen oxide ($NO_x = NO + NO_2$), coming from shipping emissions can be precursors for particle formation (Jägerbrand et al., 2019; Karl et al., 2019; Matthias et al., 2010). Sulfur dioxide is released mainly by human activities such as fossil fuel burning, petroleum refining, and metal smelting (Zhong et al., 2020). $SO_2$ is oxidized by dissolved oxidants such as $O_3$ and hydrogen peroxide ($H_2O_2$) in the aqueous phase and by OH in the gas phase to generate $H_2SO_4$ (Seinfeld and Pandis, 2006). $H_2SO_4$ and $HNO_3$ react with $NH_3$ to form ammonium sulfate (($NH_4)_2SO_4$) and $NH_4NO_3$ aerosols, with $H_2SO_4$ neutralization having preference due to its lower vapor pressure (Hauglustaine et al., 2014).

Nitrogen oxides are primarily removed during the day via the hydroxyl (OH) radical oxidation reaction to produce nitric acid ($HNO_3$; Seinfeld and Pandis, 1998). S At night, the main $NO_x$ removal method involves interacting with ozone ($O_3$) to produce the nitrate ($NO_3$) radical, which then may combine with nitrogen dioxide ($NO_2$) to form dinitrogen pentoxide ($N_2O_5$) and subsequently may undergo a heterogeneous reaction with water to produce $HNO_3$. As it is highly soluble, $HNO_3$ disperses quickly in water droplets or is neutralized by reaction with ammonia ($NH_3$) to produce $NH_4NO_3$ aerosols. Increased emissions of $NH_3$ or $HNO_3$ formation as well as their deposition negatively affect the environment through eutrophication and acidification, thereby contributing to the loss of ecosystem biodiversity (Remke et al., 2009; Kleijn et al.; 2009; Krupa, 2003). Furthermore, the air pollution status should be assessed to investigate the consequences of new legislation.

The current work investigates and analyzes the predictions of five different CTMs for air pollutant dispersion and transformation. The intercomparison was carried out in two parts: Part one included the photochemistry and differences among the models regarding $NO_2$ and $O_3$ (Fink et al., 2023). The present study is part two of the model intercomparison and evaluates the same CTM simulations but different air pollutants, namely aerosols. This paper is structured as followed: Sect. 3.1 and 3.2 considers simulated overall $PM_{2.5}$ model performance and spatial distribution. In Sect. 3.3 precursors ($NH_3$, $HNO_3$, $SO_2$ and $NO_2$) are investigated, as a base for inorganic particle species. Inorganic aerosols concentration and wet deposition is regarded in Sect. 3.4.

To date, the present study is the first multimodel study designed to compare the potential impacts of shipping on $PM_{2.5}$ and particle species simulated by five regional-scale CTMs for the Mediterranean Sea.



## 2 Materials and Methods

In this section the models participating in the intercomparison study are briefly described. More detailed information about the standard setup of models and model internal mechanisms used in the present study can be found in part 1 of this intercomparison study (Fink et al., 2023), which focuses on nitrogen oxides and ozone.

### 2.1 Models

In this study, five different regional-scale CTM systems run by four institutions participated: CAMx and CHIMERE by AtmoSud, CMAQ by Helmholtz-Zentrum Hereon, EMEP by IVL Swedish Environmental Research Institute and LOTOS-EUROS by TNO Netherlands Organization for Applied Scientific Research. For producing comparable results w.r.t the impact shipping emissions on $PM_{2.5}$ concentrations, the models were set up in a similar way. The same shipping emissions data were
used for all CTMs from STEAM (version 3.3.0.; Jalkanen et al., 2009; Jalkanen et al., 2012; Johansson et al., 2013; Johansson et al., 2017) were used for all CTMs. Land-based emissions (CAMS-REG, v2.0), grid projection (WGS84_lonlat), domain (Mediterranean Sea), grid resolution (0.1° × 0.1°, 12 × 12 km) and the modeled year (2015) were also consistent. The CTM systems were applied in their standard setup for other input data, i.e. the meteorological input data and the boundary and initial conditions differed.

The model domains covered the largest part of the Mediterranean Sea, with a spatial extent ranging in longitude from -0.95° to 29.95° and in latitude from 33.8° to 44.95° (Appendix A). The appointed grid cell size was 12 × 12 km² interpolated on a 0.1° × 0.1° grid nested in a 36 × 36 km² grid (except EMEP).

A reference run for present air quality conditions was performed using all models, including all emissions (base case). Furthermore, all models ran once without shipping emissions (noship case). The difference between the estimates with all
emissions and the calculations without shipping emissions was then used to calculate the potential impact of ships on pollutant concentrations (zero-out method).

From the results of all models, the annual averaged ensemble mean was calculated based on the daily files. The model run outputs all contained $PM_{2.5}$ in µg/m³ at a daily resolution on a 2D grid from the lowest layer and provided this as a netcdf file following CF conventions. Concentrations in the lowest layer close to ground was used for the intercomparison. The CTM
systems calculated $PM_{2.5}$ concentrations in different ways depending on the major physical and chemical mechanisms implemented. Table 1 summarizes the model setups.

The models used in the intercomparison are listed as follows:
- CAMx v6.50 (Ramboll Environ., 2016)
- CHIMERE 2017r4 (Menut et al., 2013)
- CMAQ v5.2 (Byun and Schere, 2006; Appel et al., 2017)
- EMEP MSC-W (Simpson et al., 2012; Simpson et al., 2020)
- LOTOS-EUROS v2.0 (Manders et al., 2017)



Detailed descriptions of the used models can be found in the first part of the intercomparison study (Fink et al., 2023).

**Table 1: Main model parameters and input data for the five chemical transport models.**

| Model parameter | CAMx | CHIMERE | CMAQ | EMEP | LOTOS-EUROS |
|---|---|---|---|---|---|
| **Grid resolution inner domain** | 12x12 km² | 12x12 km² | 12x12 km² | 0.1°x 0.1° | 0.1°x 0.1° |
| **Grid resolution outer domain** | 36x36 km² | 36x36 km² | 36x36 km² | none | 0.5°x 0.25° |
| **Meteorological driver** | WPS/WRF | WPS/WRF | COSMO-5 CLM | ECMWF (IFS) | ECWMF (IFS) |
| **Boundary conditions** | Mozart-4 output | Gaseous species: LMDz-INCA model<br><br>Aerosols: GOCART model | IFS_CAMS cycle45r1 | boundary conditions provided with the open source model distribution for year 2015 | CAMS C-IFS |
| **Land-based emissions** | CAMS-REG v2.2.1 | CAMS-REG v2.2.1 | CAMS-REG v2.2.1 | CAMS-REG v2.2.1 | CAMS-REG v2.2.1 |
| **Shipping emissions** | STEAM v3.3.0 | STEAM v3.3.0 | STEAM v3.3.0 | STEAM v3.3.0 | STEAM v3.3.0 |
| **Biogenic emissions** | MEGAN Model v2.03 | MEGAN Model v2.04 | MEGAN Model v3 | Calculated online | Calculated online |
| **Sea salt emissions** | Calculation based on Ovadnevaite et al. (2014) | Calculation based on Monahan et al. (1986) | Calculation based on Kelly et al. (2010) | Calculation based on Monahan et al. (1986) and Mårtensson et al. (2003) | Calculation based on Monahan et al. (1986) and Mårtensson et al. (2003) |
| **Dust emissions** | Based on approach used in global EMAC (ECHAM/ MESSy; Klingmueller et al., 2017; Astitha et al., 2012). | Calculated online | Not considered | Key parameter is wind friction velocity | Calculated online |
| **Chemical mechanism** | CB05 | MELCHIOR2 | CB05 | EmChem 19a | CBM-IV |



| Aerosol size distribution | PM$_{2.5}$; PM$_{10}$ | 8 bins: 40 nm to 10 µm | Trimodal size distribution (0.03µm, 0.3µm, 6µm; Binkowski and Roselle, 2003) | PM$_{2.5}$; PM$_{2.5-10}$ | PM$_{2.5}$; PM$_{2.5-10}$ |
|---|---|---|---|---|---|
| Inorganic aerosol module | ISORROPIA (Nenes et al., 1998) | ISORROPIA (Nenes et al., 1998) | ISORROPIA II (Fountoukis and Nenes, 2007) | MARS (Binkowski and Shankar, 1995) | ISORROPIA II (Fountoukis and Nenes, 2007) |
| Organic aerosol module | SOAP semivolatile scheme (Strader et al., 1999) | Described in Pun et al. (2006) | Updates on SOA as described in Pye et al. (2017) | For SOA the volatility basis set (VBS) approach (Robinson et al. 2007; Donahue et al. 2009; Bergström et al. 2012) is used | No organic aerosols in the simulations |
| Wet deposition scheme | Scavenging model for gases and aerosols (Seinfeld and Pandis, 1998) | The wet deposition in CHIMERE follows the scheme proposed by Loosmore & Cederwall (2004). | Wet deposition is calculated within CMAQ's cloud module as described by Roselle and Binkowsk (1999) | Calculation as described in Emberson et al. (2000); parametrization for different surfaces as in Simpson et al. (2012) | Wet deposition is divided between in-cloud and below-cloud scavenging. The in-cloud scavenging module is based on the approach described in Seinfeld and Pandis (2006) and Banzhaf et al. (2012). |
| Dry deposition scheme | Resistance model of Zhang et al. (2003) | Dry deposition is as in Wesely (1989) | Dry deposition scheme M3Dry (Pleim, 2001) | As described in Simpson et al. (2012) | Resistance approach following Erisman et al. (1994) |

### 2.1.1 Aerosol Modules

CAMx includes algorithms for inorganic aqueous chemistry (RADM-AQ), inorganic gas-aerosol partitioning (ISORROPIA), and two organic gas-aerosol partitioning and oxidation approaches (VBS or SOAP). Using gas-phase processes, these

approaches produce sulfate, nitrate, and condensable organic gases. The hybrid 1.5-D VBS is applied to provide a unified



framework for gas-aerosol partitioning and the chemical aging of both primary and secondary atmospheric organic aerosols (Ramboll Environment and Health, 2020). One crucial assumption in PSAT is that PM is allocated to the primary precursor for each type of particulate matter (i.e., $PSO_4$ is apportioned to $SO_x$ emissions, $PNO_3$ is apportioned to $NO_x$ emissions, and $PNH_4$ is apportioned to $NH_3$ emissions).

The full description of CHIMERE's inorganic and organic modules can be found in Menut et al. (2013). CHIMERE's sectional aerosol module includes emitted TPPM, secondary species such as nitrate, sulfate, ammonium, and SOAs. Natural dust and sea salt aerosols can also be produced as passive tracers or interactive species in equilibrium with other ions. Organic matter and elemental carbon can be speciated if an inventory of their emissions is supplied. The utilized models include the aqueous, gaseous, and particulate phases of ammonia, ammonium, nitrate, and sulfate. For instance, in accordance with the ISORROPIA

thermodynamic equilibrium model, the model species $pNH_3$ represents an equivalent ammonium in the particulate phase as the sum of the $NH_4^+$ ion, $NH_3$ liquid, $NH_4NO_3$ solid, and other salts (Nenes et al., 1998).

CMAQ represents aerosol formation and growth using three log-normal distributed modes: the Aitken and accumulation modes are generally less than 2.5 μm in diameter, while the coarse mode contains significant amounts of mass above 2.5 μm. $PM_{2.5}$ and $PM_{10}$ can be obtained from the model-predicted mass concentration and size distribution information.

The CMAQ aerosol scheme AERO6 was employed; this scheme expands the chemical speciation of PM by the species Al, Ca, Fe, Si, Ti, Mg, K, and Mn. Sulfuric acid ($H_2SO_4$), nitric acid ($HNO_3$), hydrochloric acid (HCl) and ammonia ($NH_3$) gas phase – aerosol partition equilibrium is solved by the ISORROPIA II mechanism (Fountoukis and Nenes, 2007; Nenes et al., 1998). Contained within this scheme is the formation of SOA from isoprene, terpenes, benzene, toluene, xylene and alkanes (Carlton et al., 2010; Pye and Pouliot, 2012). CMAQ allows for dynamic mass transfer of semi-volatile inorganic gases to

coarse mode particles, which facilitates the replacement of chloride by $NO_3^-$ in sea salt aerosols (Foley et al., 2010).

The EMEP MSC-W model version used was rv4.34 with chemical mechanism EmChem 19a (Simpson et al. 2012; Simpson et al. 2020). The mechanism builds on surrogate VOC species (as in Simpson et al. 2012, but extended with benzene and toluene) and has 171 gas-phase and heterogeneous reactions. The model always assumes equilibrium between the gas and aerosol phases using the MARS equilibrium module of Binkowski and Shankar (1995). For SOAs a VBS approach is used

(Robinson et al. 2007; Donahue et al. 2009; Bergström et al. 2012). The semivolatile ASOA and BSOA species are considered to oxidize (age) in the atmosphere via OH reactions, whereas all POA emissions are treated as nonvolatile to maintain the emission totals of both the PM and VOC components from the official emission inventories (Simpson et al., 2012). The aerosol module of the EMEP model distinguishes five classes of fine and coarse particles (fine-mode nitrate and ammonium, other fine-mode particles, coarse nitrate, coarse sea-salt, and coarse dust); for dry-deposition purposes, these particles are assigned

mass-median diameters ($D_p$), geometric standard deviations ($\sigma_g$), and densities ($\rho_p$). The aerosol components that are taken into account include sea salt, $SO_4^{2-}$, $NO_3^-$, $NH_4^+$, and anthropogenic main PM. Aerosol water is also considered.

LOTOS-EUROS uses the TNO CBM-IV scheme, which is a modified version of the original CBM-IV scheme (Whitten et al., 1980). $N_2O_5$ hydrolysis is described explicitly based on the available (wet) aerosol surface area (Schaap et al., 2004). The



aqueous phase and heterogeneous formation of sulfate is described by a simple first-order reaction constant (Schaap et al.,
2004; Barbu et al., 2009). Aerosol chemistry is represented using ISORROPIA II (Fountoukis and Nenes, 2007).

### 2.1.2 Wet Deposition Mechanisms

Wet deposition is the predominant removal process for fine particles. The CAMx wet deposition model uses a scavenging method in which the local concentration change rate inside or under a precipitating cloud is determined by a scavenging coefficient. From the top of the precipitation profile to the surface, wet scavenging is estimated for each layer inside a
precipitating grid column. The scavenging coefficients of gases and PM are calculated differently depending on the correlations given by Seinfeld and Pandis (2006) (Ramboll Environment and Health, 2020). The wet deposition process in CHIMERE follows the scheme proposed by Loosmore & Cederwall (2004). In CMAQ, wet deposition is calculated in cloud chemistry treatments. The resolved cloud model calculates the contribution of each model layer to the precipitation. Based on a normalized profile of precipitating hydrometeors, CMAQ operates a simple algorithm to assign precipitation amounts to
individual layers (Foley et al., 2010). In EMEP, wet scavenging is treated with simple scavenging ratios, taking into account in-cloud and sub-cloud processes. The EMEP model's parameterization of wet deposition processes covers both the in-cloud and sub-cloud scavenging of gases and particles. The parameterization of wet deposition is described in Berge and Jakobsen (1998). There are two types of wet deposition in LOTOS-EUROS: below-cloud scavenging and in-cloud scavenging. The technique is described in Seinfeld and Pandis (2006), and Banzhaf et al. (2012) served as the foundation for the in-cloud
scavenging module.

### 2.2 Emissions

### 2.2.1 Land-based Emissions

All five models used anthropogenic land-based gridded emissions from the CAMs-REG v2.2 emission inventory for 2015, which is described in Granier et al. (2019) and essentially a further development of the earlier TNO_MACC inventories
(Kuenen et al., 2014). A more recent version CAMS-REG-v4.2 is described in detail in Kuenen et al. (2022).
For each country, the gridded emission files included GNFR emission sectors for the air pollutants $NO_x$, $SO_2$, NMVOC, $NH_3$, CO, $PM_{10}$, $PM_{2.5}$, and $CH_4$. The spatial resolution of the emissions data was $1/10° \times 1/20°$ in longitude and latitude (i.e., ~ 6 × 6 km over central Europe). The CAMS-REG inventory also provides default information to apply the emissions in the CTMs. The height distribution of emissions per GNFR sector was prepared according to Bieser et al. (2011). Based on assignment of
PM and NMVOC components at a detailed subsector level, PM and NMVOC speciation profiles are provided for each country, year and GNFR sector. The temporal distribution of emissions is based on the default temporal variation provided along with the CAMS-REG inventory. The NOx splitting was performed according to Manders-Groot et al. (2016).



### 2.2.2 Shipping Emissions

The shipping emission dataset produced with the STEAM model has a spatial resolution of $12 \times 12$ km² and a temporal
resolution of one hour. The STEAM v3.3.0 emissions are divided into two vertical layers (0 m to 36 m; 36 m to 1000 m) and
are provided for mineral ash, carbon monoxide (CO), carbon dioxide ($CO_2$), elemental carbon (EC), $NO_x$, organic carbon (OC),
$PM_{2.5}$, particle number count (PNC), sulfate ($SO_4$), $SO_x$ (containing $SO_2$ and $SO_3$) and VOC. To reduce the number of generated
emission maps and the computational resources needed to run the STEAM model, VOC emissions were divided into four
categories according to their properties as a function of the engine load. Emission factors for VOC are based on the average
values taken from various publications (Agrawal et al., 2008; Agrawal et al., 2010; Sippula et al., 2014; Reichle et al., 2015).

All shipping emissions are included in the lowest layer of CAMx. In CAMx, all gridded emissions are at the ground level
except punctual and linear emissions. For CHIMERE, 88 % of the emissions below 36 m and all shipping emissions above 36
m were added to the second layer. Only 12 % of the emissions below 36 m were allocated to the model's lowest layer. The
STEAM emission dataset, which included stack heights, was used for this procedure. In CMAQ, shipping emissions were split
between the two lowest levels; those below 36 m were ascribed to the lowest layer, while those above 36 m were positioned
in the second layer. The height of the lowest layer in CMAQ are 42 m for each. The STEAM emissions were summed from
hourly to daily emissions and attributed to the lowest layer (up to 90 m) in the EMEP simulations. In LOTOS-EUROS,
emissions below 36 m were divided into two layers: the first layer was 25 m thick (~ 70 % of emissions), and the second layer
was 30 m thick (~ 30 % of emissions). Over 36 m, emissions are separated into various height groups: 30 % were between 36
m and 90 m, 30% were between 170 m and 90 m, 30 % were between 170 m and 310 m, and 10 % were between 310 m and
470 m. These emissions were placed in the second or third model layers because of the dynamic second model layer, which
follows the meteorological boundary layer. All emissions were placed in this second layer when the meteorological boundary
layer was well mixed and vertically extended (higher than 470 m), while some emissions were placed in the third layer when
the boundary layer was shallow.

### 2.3 Observational Data, Statistical Analysis and Analysis of Model Results

The model findings regarding the total surface $PM_{2.5}$ concentrations from the five CTM systems were compared to data from
the   air   quality   monitoring   network   obtained   from   the   EEA   download   service
(https://discomap.eea.europa.eu/map/fme/AirQualityExport.htm, 2021). The locations of the measurement stations are shown
in Figure A1, and full information on the stations can be found in Appendix B.
The stations were chosen based on the following criteria: i) the station type was "background," ii) the station elevation was
less than 1000 meters, and iii) the station recorded data for more than one of the following pollutants: $NO_2$, $O_3$, or $PM_{2.5}$. In
the first part of this intercomparison study (Fink et al., 2023), $NO_2$ and $O_3$, were discussed. Since simulating the potential
impact of ships was the main focus of this study, stations near the sea were preferably chosen.



The model findings regarding the total surface PM$_{2.5}$ concentrations from the five CTM systems were compared to existing

observations. The RMSE, NMB, and correlation coefficient R were determined for each monitoring station to quantify the model performance, as described in the previous study (Fink et al., 2013).

A categorization scheme for the correlations was established as described in Schober et al. (2018), with weak (0.00-0.39), moderate (0.40-0.69) and strong (0.70-1.00) correlations.

To compare the predicted daily mean concentrations to the measurements recorded at representative sites, time series were

employed. In addition, based on hourly data, the yearly mean potential ship impact was determined. Boxplots based on yearly values obtained from hourly data at each station were used to graphically compare the model performances using the R, NMB, and RMSE metrics. Annual mean values based on hourly data were utilized for the intercomparison maps. Based on hourly data, the correlations between models were determined for each grid cell.



## 3 Results

### 3.1 PM$_{2.5}$ Model Performance

Regarding the model performance, time series can give an overview of the performance throughout the whole year. Figure 1 displays the average values at all 28 measurement stations. CAMx, CMAQ, EMEP and LOTOS-EUROS underestimate the actual measured data. The largest underestimations are found for CMAQ (NMB = -0.42) and LOTOS-EUROS (NMB = -0.54). However, the correlations between the modeled and measured data is strongest for these models (CMAQ: R = 0.50, LOTOS-

EUROS: R = 0.54; Table 2). No correlation can be found between the measured and modeled data for CHIMERE (R = 0.02), on the other hand CHIMERE displays only a slight overestimation of the actual data (NMB = 0.06). The simulated potential impacts of ships at all measurement stations are between 5.7 % (CMAQ) and 13.8 % (CAMx; Table 2) as annual average. The simulated ship impacts on PM$_{2.5}$ concentration are within the ranges stated in other studies. In a review of studies regarding the impact of shipping emissions on coastal regions, Viana et al. (2014) reported PM$_{2.5}$ impacts of shipping between 5 % and

14 %. Aksoyoglu et al. (2016) found PM$_{2.5}$ concentrations between 10 % and 15 % along coastal areas due to ship traffic. Ship impacts of approximately 20 % in the southern coastal region of the Iberian Peninsula were found by Nunes et al. (2020). Although in this study, the utilized models underestimated the actual measured total PM$_{2.5}$ concentrations, they slightly overestimated the relative potential ship impact on PM$_{2.5}$ compared to previous measurement studies. Donateo et al. (2014) measured a proportion of 7.4 % of ships to total PM$_{2.5}$; Pandolfi et al. (2011) measured a proportion of shipping in the bay of

Algeciras to PM$_{2.5}$ concentrations between 5 % and 10 %. Argawal et al. (2009) monitored PM$_{2.5}$ at the harbor of Los Angeles and found PM$_{2.5}$ contributions from ships up to 8.8 %. Predominating secondary particles in PM$_{2.5}$ for potential ship impact in the present study can explain the deviations to the measurement studies.

The RMSE is very similar for all models with a value between 10.7 µg/m³ and 12.2 µg/m³. However, the RMSE is strongly determined by high concentrations and can be biased by outliers. This might explain the similar RMSE derived from

CHIMERE despite the lack of correlation. The mean RMSE from different models for PM$_{2.5}$ in Europe found in the AQMEII intercomparison study by Im et al. (2015) was 6.19 for rural stations and 10.26 for urban stations and is similar as the RMSE calculated in the present study.

The underestimation of PM$_{2.5}$ concentrations by four out of five models is consistent with results by Im et al. (2015) who reported an underestimation of particulate matter for all participating models, with largest underestimations observed in the

Mediterranean region. They stated that the representation of dust and sea-salt emissions had a large impact on the simulated PM concentrations and that uncertainties remain when trying to identify the reasons for the model bias (Im et al., 2015). Additionally, in a study by Gašparac et al. (2020), underestimations were also found when using EMEP and WRF-Chem to model PM$_{2.5}$ at rural stations in Europe. Solazzo et al. (2012) performed an operational model evaluation for ten models and found that the models underestimated the monthly mean PM$_{2.5}$ surface concentrations in Europe in most cases.




**Table 2: Correlation (R), normalized mean bias (NMB), root mean square error (RMSE), observational (obs) and modeled (mod) mean PM$_{2.5}$ values for 2015 over all 28 stations. Observed mean value for all stations is 14.6 µg/m³.**

| | Correlation | NMB | RMSE (µg/m³) | Mod (µg/m³) | Absolute potential ship impact (annual mean average at all stations) in µg/m³ | Relative potential ship impact (annual mean average at all stations) in % |
|---|---|---|---|---|---|---|
| **CAMx** | 0.19 | -0.33 | 11.5 | 8.9 | 1.2 | 13.8 |
| **CHIMERE** | 0.02 | 0.06 | 11.1 | 14.3 | 1.8 | 13.2 |
| **CMAQ** | 0.50 | -0.42 | 10.7 | 8.3 | 0.5 | 5.7 |
| **EMEP** | 0.17 | -0.33 | 12.2 | 8.9 | 0.9 | 9.1 |
| **LOTOS-EUROS** | 0.54 | -0.53 | 10.9 | 6.8 | 0.6 | 9.5 |






**Figure 1: Time series with daily mean PM$_{2.5}$ concentration in 2015, averaged for all stations and the respective grid cells of the models. (a) = CAMx, (b) = CHIMERE, (c) = CMAQ, (d) = EMEP, (e) = LOTOS-EUROS. Dashed gray line = measured data, colored lines = modelled data, gray line = modelled potential ship impact.**



## 3.2 PM$_{2.5}$ Spatial Distribution

The highest PM$_{2.5}$ values are simulated by all five models in northern Italy, the Balkan Peninsula and northern Africa (Figure 2). The PM$_{2.5}$ annual mean concentration results show that CHIMERE has the highest annual mean values of 13 µg/m³ to 15 µg/m³ for the eastern part of the domain and over water, whereas LOTOS-EUROS displays the lowest values with 2.0 µg/m³ to 4.0 µg/m³ in most regions (Figure 2). CMAQ, CAMx and EMEP show similar model PM$_{2.5}$ outputs with diverse values distributed between 2.0 µg/m³ and 11 µg/m³ over the domain. The ensemble mean value over the whole domain is 8.6 µg/m³ (Figure 8 a). All five models display high PM$_{2.5}$ concentrations of >15 µg/m³ in the Po valley. In this area, Kiesewetter et al. (2015) and Clappier et al. (2021) also simulated high values between 20 µg/m³ and 45 µg/m³ for 2015. As demonstrated in Table 3, the correlation between the base-run model results with all emissions is strongest between EMEP and CMAQ (R = 0.59) and CAMx and CMAQ (R = 0.42). In Fink et al. (2023) a high correlation was found between CAMx and CHIMERE simulated NO$_2$ and O$_3$ concentration because both models used the same meteorology. Nevertheless, the present study reveals that particle chemistry causes more differing results due to a higher complexity in the calculations.

The potential impacts of PM$_{2.5}$ from ships simulated by CAMx, LOTOS-EUROS and EMEP have the largest areas with values up to 25 % at the main shipping routes (Figure 3). CMAQ and CHIMERE have a potential shipping impact of 15 % along the main shipping lines close to the African coast. This impact is lower than that shown in other studies. Aksoyoglu et al. (2016) found the highest impacts of 25 % to 50 % of total PM$_{2.5}$ concentrations when using CAMx along main shipping routes. Sotiropoulou & Tagaris (2017) used CMAQ for simulations and stated that emissions from shipping are likely to increase PM$_{2.5}$ concentrations during winter by up to 40 % over the Mediterranean Sea, while during summer, they simulated an increase of more than 50 %. In both studies, the modeled year is 2006, which might explain the deviation to the present study using a different year. Regarding coastal areas in the present study, potential shipping impacts reaching to 12 % to 15 % are simulated. Regarding the absolute potential impacts of ships at the main shipping routes, CAMx, CHIMERE and EMEP show values of 2.0 µg/m³, and the values simulated by CMAQ and LOTOS-EUROS are between 0.5 µg/m³ and 1.0 µg/m³ (Figure 4). The median of the ensemble mean is 0.85 µg/m³ (Figure 4 and 8). Aksoyoglu et al. (2016) simulated similar shipping impacts with CAMx, with values mainly between 0.5 µg/m³ and 1.0 µg/m³.

The annual mean sea salt (NaCl) concentration in fine and coarse showed the highest values for CHIMERE, which might be an explanation for the high PM$_{2.5}$ absolute concentration (Figure S1). In contrast, the LOTOS-EUROS sea salt concentration displayed similar values as CHIMERE, but the overall PM$_{2.5}$ concentration is lowest compared to the other CTMs. CAMx, CMAQ and EMEP showed annual mean sea salt concentrations between 0.0 µg/m³ and 1.0 µg/m³.

Solazzo et al. (2012) demonstrated that the chemical components SO$_4^{2-}$, NO$_3^-$ and NH$_4^+$ were better reproduced by nine CTMs than total PM$_{2.5}$. They concluded from this result that other components (e.g., organic aerosols) could be simulated with less accuracy than inorganic components.





**Table 3: Correlations between models for the PM$_{2.5}$ base runs of the whole domain (all grid cells), based on daily PM$_{2.5}$ total concentration data.**

| All | CAMx | CHIMERE | CMAQ | EMEP | LOTOS-EUROS |
|---|---|---|---|---|---|
| **LOTOS-EUROS** | 0.07 | 0.00 | 0.26 | 0.06 | - |
| **EMEP** | 0.32 | 0.17 | 0.59 | - | |
| **CMAQ** | 0.42 | 0.19 | - | | |
| **CHIMERE** | 0.40 | - | | | |
| **CAMx** | - | | | | |






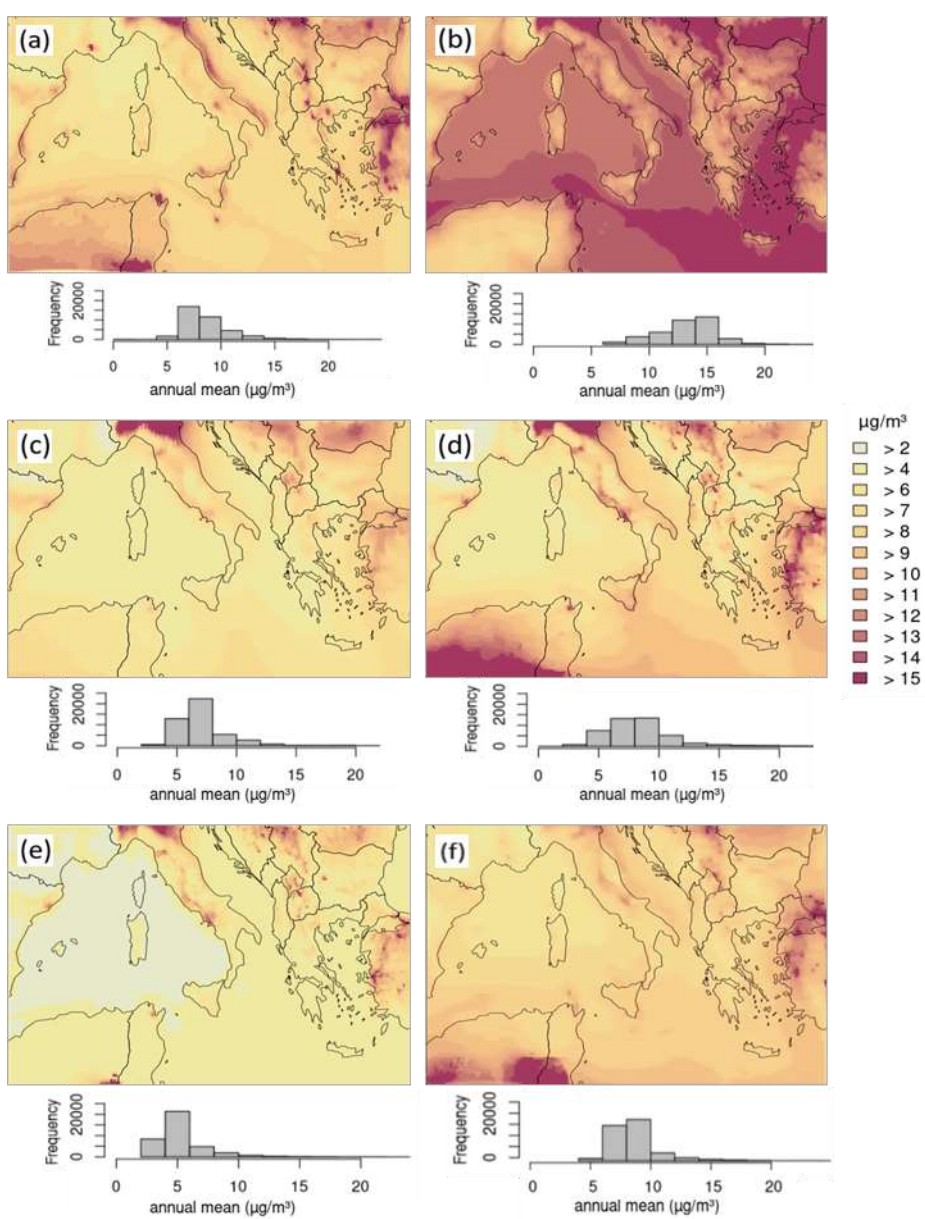

**Figure 2: Annual mean PM$_{2.5}$ total concentration. (a) = CAMx, (b) = CHIMERE, (c) = CMAQ, (d) = EMEP, (e) = LOTOS-EUROS, (f) = ensemble model mean. Below the domain figure is the respective frequency distribution displayed for the annual mean PM$_{2.5}$ concentration, referred to the whole model domain.**



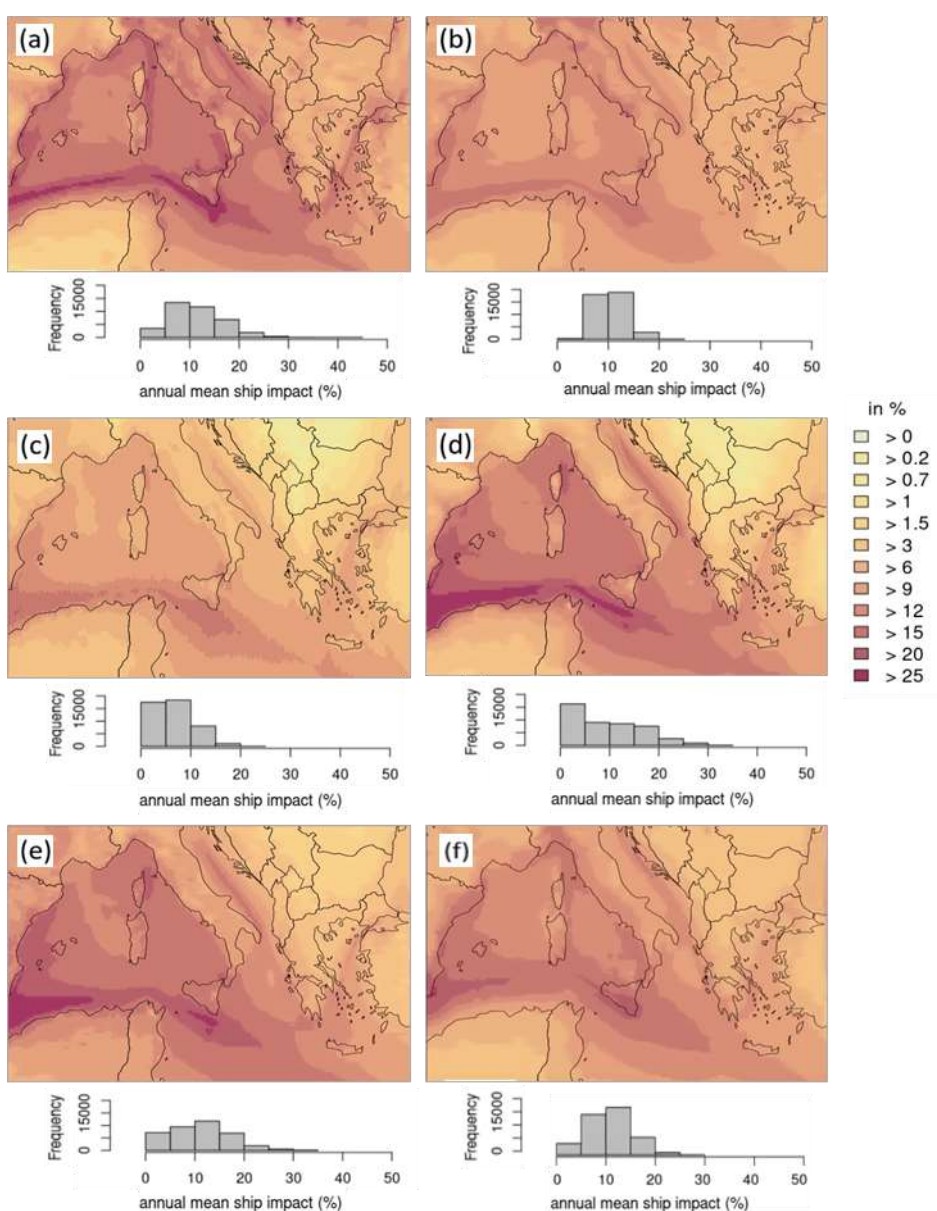

**Figure 3: Annual mean PM$_{2.5}$ relative potential ship impact. (a) = CAMx, (b) = CHIMERE, (c) = CMAQ, (d) = EMEP, (e) = LOTOS-EUROS, (f) = ensemble model mean. Below the domain figure is the respective frequency distribution displayed for the annual mean PM$_{2.5}$ potential ship impact, referred to the whole model domain.**




**Figure 4: Annual mean PM$_{2.5}$ absolute potential ship impact. (a) = CAMx, (b) = CHIMERE, (c) = CMAQ, (d) = EMEP, (e) = LOTOS-EUROS, (f) = ensemble model mean. Below the domain figure is the respective frequency distribution displayed for the annual mean PM$_{2.5}$ potential ship impact, referred to the whole model domain.**



### 3.3 Precursors

High amounts of $NH_3$, $HNO_3$, $SO_2$ and $NO_2$ are expected to lead to higher values of the aerosol particles composed of $NH_4^+$, $NO_3^-$ and $SO_4^{2-}$. The modelled spatial distributions of these precursors can be found in the Supplements ($HNO_3$: Figures S2-

S4; $NH_3$: Figures S6-S8; $SO_2$: Figures S9-S11; and $NO_x$: Figures S12-S14).

The highest annual mean $HNO_3$ concentration among the base runs is found in the CAMx and the CHIMERE simulations over water (2.0 µg/m³ to 5.0 µg/m³); over land, the values are between 0.0 µg/m³ and 1.5 µg/m³, and those in coastal areas reached 2.0 µg/m³ (Figure S2). The absolute potential ship impact is also highest in CAMx and CHIMERE at the main shipping routes and over water areas (1.0 µg/m³ to 3.0 µg/m³). The relative potential ship impact on total $HNO_3$ ranges from 60 % to 85 %

along the main shipping routes simulated by CAMx, CMAQ and EMEP (Figure S3). These impacts are slightly lower for CHIMERE and LOTOS-EUROS (60 % to 75 %).

The high $HNO_3$ concentrations simulated by CAMx and CHIMERE might be traced back to the $NO_2$ concentrations; these two models also show higher $NO_2$ concentrations than the other CTMs (Figure S12; Fink et al., 2023). This can be explained by the fact that $HNO_3$ is a major $NO_2$ sink, especially during daytime. $NO_2$ is primarily emitted from anthropogenic fossil fuel

burning but also comes from natural sources (i.e., soil emissions, biomass burning, lightning). During daytime, the main $NO_2$ removal mechanism is oxidation by hydroxyl (OH) radicals to form $HNO_3$ (Seinfeld and Pandis, 1998). It can be concluded that in areas with shipping, more $NO_2$ enters the atmosphere, the total $NO_2$ concentration increases, and as a result of the subsequent reactions, the $HNO_3$ concentration also increases. The $HNO_3$:$NO_2$ ratio can be used to normalize the data (Figure S5). The ratio displays low values over land and along main shipping routes, indicating that in these areas, both

the $HNO_3$ and $NO_2$ concentrations are high. Low $HNO_3$:$NO_2$ ratio could also mean that only a small amount of OH is present, especially in areas with low $O_3$ concentration.

After its formation, $HNO_3$ can react with $NH_3$ to be neutralized and form particles when $NH_3$ is in excess. The annual mean $NH_3$ for the base case show very similar patterns and values among all models (Figure S6). The highest concentrations of $NH_3$ with all emission sources are located over land areas with values up to 2.5 µg/m³, which can be traced back to agriculture, the

main source of $NH_3$ emissions (Behera et al., 2013). Over water areas, the $NH_3$ concentration is very small, typically between 0.0 µg/m³ and 0.3 µg/m³, except for the slightly higher results modeled by LOTOS-EUROS, with values between 0.2 µg/m³ and 0.8 µg/m³. Negative potential ship impacts (-0.01 µg/m³ to -1.0 µg/m³ and -2.5 % to -150 %; Figure S7 and S8) are found for the whole domain in all five models. The relative ship impacts are lowest at the main shipping routes for CAMx and EMEP. The spatial distribution of the $NH_3$ relative ship impact is opposite to the simulated $HNO_3$ values; at the main shipping routes

with low $NH_3$ and high $HNO_3$ values. These results indicate that available $NH_3$ reacts directly with $HNO_3$ to form particles (i.e., $NH_4NO_3$). Thus, $NO_x$ emissions from shipping lead to $HNO_3$ formations and subsequent $NH_3$ consumption, e.g. shipping impacts on $NH_3$ concentrations are usually negative.

The CAMx simulations show highest $SO_2$ concentrations with more than 10 µg/m³ in some areas in Western Turkey, in urban areas and along major shipping lanes (Figure S9). The results from the other four CTMs display high values around the



Bosporus and in some areas over the Balkan Peninsula with values of 11 µg/m³ and much lower concentrations along the main shipping routes. The potential ship impacts are similarly high in CAMx and CHIMERE (1.0 µg/m³; 85 % to total concentration; Figure S10 and S11), with the highest values along the major shipping route north of the African coast. The CMAQ, EMEP and LOTOS-EUROS results display similarly high values but only in small areas. The modeled year is 2015, so the global 0.5 % sulfur cap of marine fuels was not yet effective. Heavy fuel oils with sulfur contents reaching 3.50 % were used until 2020

to power ships; thus, the $SO_2$ emitted from ships in the present study is still high and it can be expected that it has a large impact on secondary particle formation.

### 3.4 Inorganic Aerosol Species

### 3.4.1 Concentrations

In the Northern Hemisphere, secondary inorganic ammonium, sulfate and nitrate aerosols represent a large fraction of the

$PM_{2.5}$ composition (Jimenez et al., 2009). Ammonium preferentially binds to $SO_4^{2-}$ in atmospheric aerosols in the form of $(NH_4)_2SO_4$. $NH_4NO_3$, on the other hand, is formed in areas characterized by high $NH_3$ and $HNO_3$ conditions and low $H_2SO_4$ conditions. The results of the CTMs with regard to these three particle species and their potential ship impacts are considered in the following section. The spatial distributions of the total concentrations and absolute potential ship impacts of the individual species can be found in the Supplements ($NH_4^+$: Figures S15 & S16; $SO_4^{2-}$: Figures S17 & S18; and $NO_3^-$: Figures

S19 & S20), spatial distribution of relative potential ship impact is shown in Fig. 5 to Fig. 7.

The spatial distribution of $NH_4^+$ shows that the lowest total annual mean can be found mainly in the southwestern part of the domain (approximately 0.0 µg/m³) and the highest in the Po Valley and Bosporus (1.5 µg/m³, Figure S15). The relative ship impacts are very similar for all models (0.25 % to 5.0 % over land, 10 % to 25 % over water; Figure 5) as well as for the absolute ship impact (Figure S16). Aksoyoglu et al. (2016) simulated $NH_4^+$ values between 0.0 µg/m³ and 0.2 µg/m³ in the

Mediterranean region, with higher concentrations (0.4 µg/m³) in the Po valley. This is within the same range of concentrations in the present study. Ge et al. (2021) used the EMEP model to simulate global particle species concentrations and compared them to measured concentrations. They showed in their study that the $NH_4^+$ concentrations simulated in Europe in 2015 were overestimated by factor 2 compared to the actual measured $NH_4^+$ concentrations. The measurements displayed a mean of 0.45 µg/m³. The ensemble mean for $NH_4^+$ in the present study (0.6 µg/m³, Figure 8 a) is in good agreement with these measurements.

However, a previous study on measured compared with simulated aerosol distribution with the CMAQ model displayed a slight underestimation of $NH_4^+$ (Matthias, 2008).

The $NH_4^+$ proportion to total $PM_{2.5}$ is similar among all models (5.6 % to 7.8 %; Figure 8 a, Table 4), and only LOTOS-EUROS displayed a relatively high share (12.2 %). This pattern is similar for the ship impacts, where all models show proportions between 9.1 % and 12.6 %, but higher values are simulated by LOTOS-EUROS (23.5 %; Figure 8 b, Table 5).

$SO_4^{2-}$ is the oxidation product of $SO_2$, which is primarily emitted by anthropogenic processes such as fossil fuel combustion, petroleum refining, and metal smelting (Zhong et al., 2020). In the present study, $SO_4^{2-}$ is the main contributor to total $PM_{2.5}$



mass (Figure 8, Table 4). Especially in the model ensemble mean for the absolute ship-related concentrations, $SO_4^{2-}$ makes up 44.6 % of $PM_{2.5}$ (Figure 8 b, Table 5). The annual mean $SO_4^{2-}$ total concentration is highest for CHIMERE in the eastern part of the domain, reaching 6.0 µg/m². EMEP displays a $SO_4^{2-}$ concentration within the ranges of the other models CAMx, CMAQ

and LOTOS-EUROS in the western part of the domain. These models show very similar spatial distributions with concentrations up to 2.0 µg/m³. The median ensemble mean for the run with all emission sources is 2.0 µg/m³. This ensemble mean is low in comparison with the results of Solazzo et al. (2012); they found a mean value of 6.0 µg/m³ but considered a larger European area that included the areas with highest $SO_4^{2-}$ concentrations in Europe. For this larger area, Solazzo et al. (2012) found that the used models underestimated $SO_4^{2-}$ by 7 % to 17 %.

In the present study, the relative potential ship impact on total $SO_4^{2-}$ is lowest over land, with 0 % to 3.0 %, and higher in coastal areas, with values from 6 % to 20 % (Figure 6). Along the main shipping routes it is highest, reaching 50 % for CAMx, EMEP and LOTOS; for CHIMERE and CMAQ, it is lower with values reaching 30 %. Aksoyoglu et al. (2016) showed similar relative potential ship impacts of 50 % to 60 % in the western Mediterranean. In their study, values were between 0.0 µg/m³ to 1.0 µg/m³ over land areas, but over water along the main shipping routes they were highest at 2.2 µg/m³.

In the present study, CTM systems simulated lower values for ship impacts; over land, they are 0.0 µg/m³ to 0.03 µg/m³, and along the main shipping routes, they reached 0.9 µg/m³. Regarding the absolute ship impacts on $SO_4^{2-}$, the model simulations displayed similar concentrations and are slightly lower for CMAQ and LOTOS-EUROS (Figure S18) compared to the other models. Especially over water areas, large areas with considerable $SO_2$ and $SO_4^{2-}$ concentrations can be seen. Because $NH_4^+$ is preferably bound to $SO_4^{2-}$ in atmospheric aerosols to form $(NH_4)_2SO_4$, in areas over water, less $NH_4NO_3$ forms.

Im et al. (2014) suggested in their intercomparison study that over Europe, $SO_4^{2-}$ levels were underestimated by most models; only a few models overestimated $SO_4^{2-}$ concentrations in Europe. The underestimating models were WRF-CHEM models, and the $SO_4^{2-}$ underestimations were attributed to the absence of $SO_2$ oxidation in cloud water in the heterogeneous phase.

The highest annual mean $NO_3^-$ total concentrations is simulated over land areas especially over Italy and in the Balkan states (> 2 µg/m³; Figure S19), lowest concentration are over sea. CAMx, CMAQ and LOTOS-EUROS show higher concentrations

compared to results derived from CHIMERE. The concentrations over water are lower than those over land. The ensemble median of all CTMs over the whole domain is 0.63 µg/m³ (median value; Figure 8 a). The absolute potential impacts of ships on the total $NO_3^-$ concentrations are similar among all models, displaying values mainly between -0.005 µg/m³ and 0.15 µg/m³; only CMAQ demonstrates relatively low values along the main shipping routes (-0.5 µg/m³), and CAMx has higher values (1.0 µg/m³) in some coastal areas (Figure S20). This can be explained by higher $SO_4^{2-}$ concentrations derived from $SO_2$

emissions. Sulfate replaces nitrate as long as ammonia concentration is low. In model simulations with ships, $NO_3^-$ can decrease because ammonia is already taken from sulfur emissions from ships. Aksoyoglu et al. (2016) found similar results for the Mediterranean Sea considering the $NO_3^-$ concentrations, with values between 0.0 µg/m³ and 0.2 µg/m³. Im et al. (2014) showed that simulated $NO_3^-$ levels were overestimated by most of the CTMs by more than 75 %. Higher concentration over water than over land due to $NH_4NO_3$ formation are found in areas characterized by high $NH_3$ and $HNO_3$ conditions and low $H_2SO_4$

conditions. In the present study, the relative potential ship impacts on $NO_3^-$ displays contradicting tendencies among the models



(Figure 7). The CAMx, EMEP and LOTOS model results are similar, with relative potential ship impacts over land of 0.0 % to 5.0 % (in the Balkan states), those in coastal areas and Italy of 10 % to 25 % and those along main shipping routes of 50 % to 65 % or even up to 85 %. CHIMERE and CMAQ display lower relative potential ship impacts. For CMAQ, the impact is even negative along the main shipping routes, at -25 %. Sulfur dioxide or ammonia, might lead to negative $NO_3^-$ impact,

because the $NO_2$ emissions from ships would make a positive contribution to nitrate formation. Therefore, without ships, a $(NH_4)_2SO_4$ should be formed, which is more stable than $NH_4NO_3$. These low values in the aerosol species for CMAQ but higher values for EMEP, CAMx and LOTOS represented the $PM_{2.5}$ ship impacts and might partly explain the deviations in $PM_{2.5}$. Furthermore, in CMAQ the coarse mode in nitrate and ammonium has a larger share compared to the other CTMs. A more detailed discussion will be given in Sect. 4.

Regarding the $PM_{2.5}$ composition, the share of other particles, which contain mainly organics but also e.g. sea salt, is highest compared to the inorganic species (Figure 8). Nevertheless, the particle composition revealed varying distributions in the ship-related $PM_{2.5}$ concentration. Here, inorganic particle species have relatively high percentages compared to organic aerosols. In some cases, sulfate has an even higher share of the total $PM_{2.5}$ than other particles.

The seasonal variability in particle species shows that $NO_3^-$ is more temperature-dependent than $SO_4^{2-}$ and $NH_4^+$. $NO_3^-$ is higher

in winter and spring but lower in summer and autumn. This pattern can be found in all CTM simulations. For $PM_{2.5}$, on the other hand, no discernible pattern is found regarding seasonal variability. In particular, the ensemble mean $PM_{2.5}$ concentration remained within the same range in all seasons.





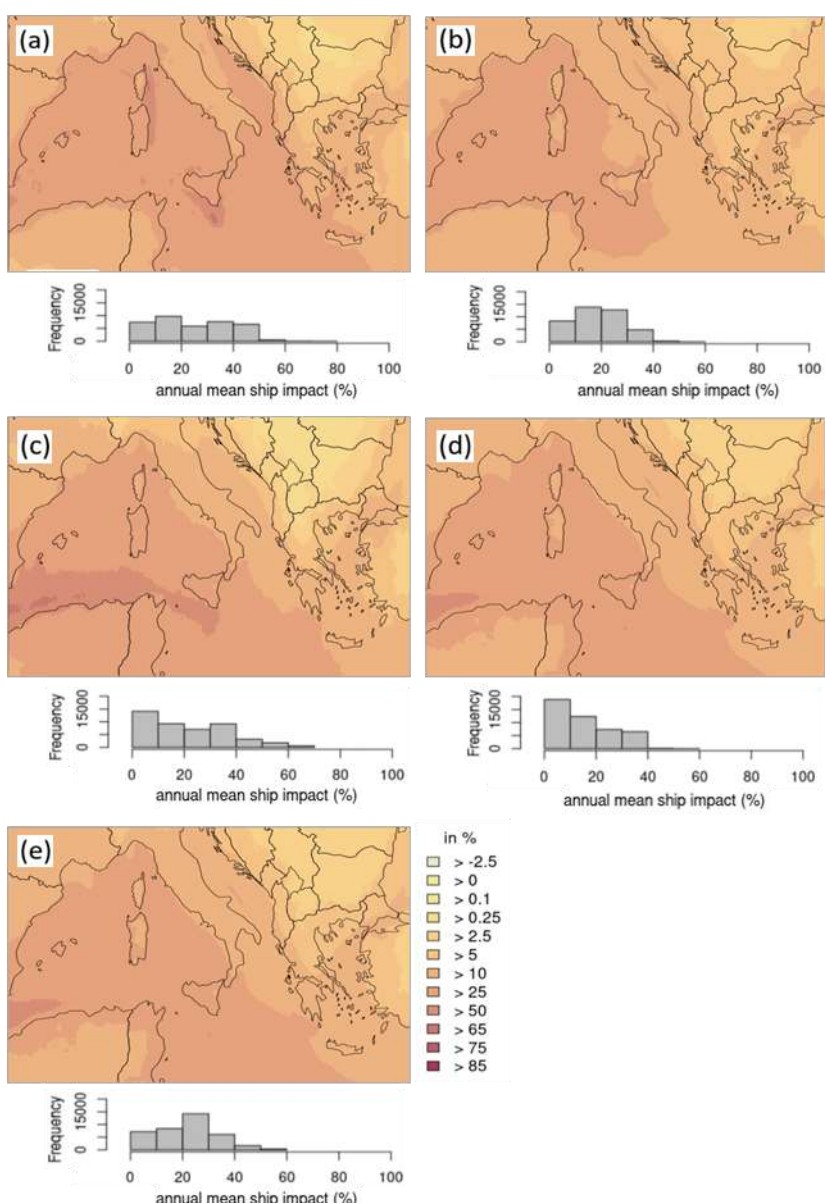

**Figure 5: Annual mean NH$_4^+$ relative potential ship impact. (a) = CAMx, (b) = CHIMERE, (c) = CMAQ, (d) = EMEP, (e) = LOTOS-EUROS. Below the domain figure is the respective frequency distribution displayed for the annual mean NH$_4^+$ potential ship impact, referred to the whole model domain.**






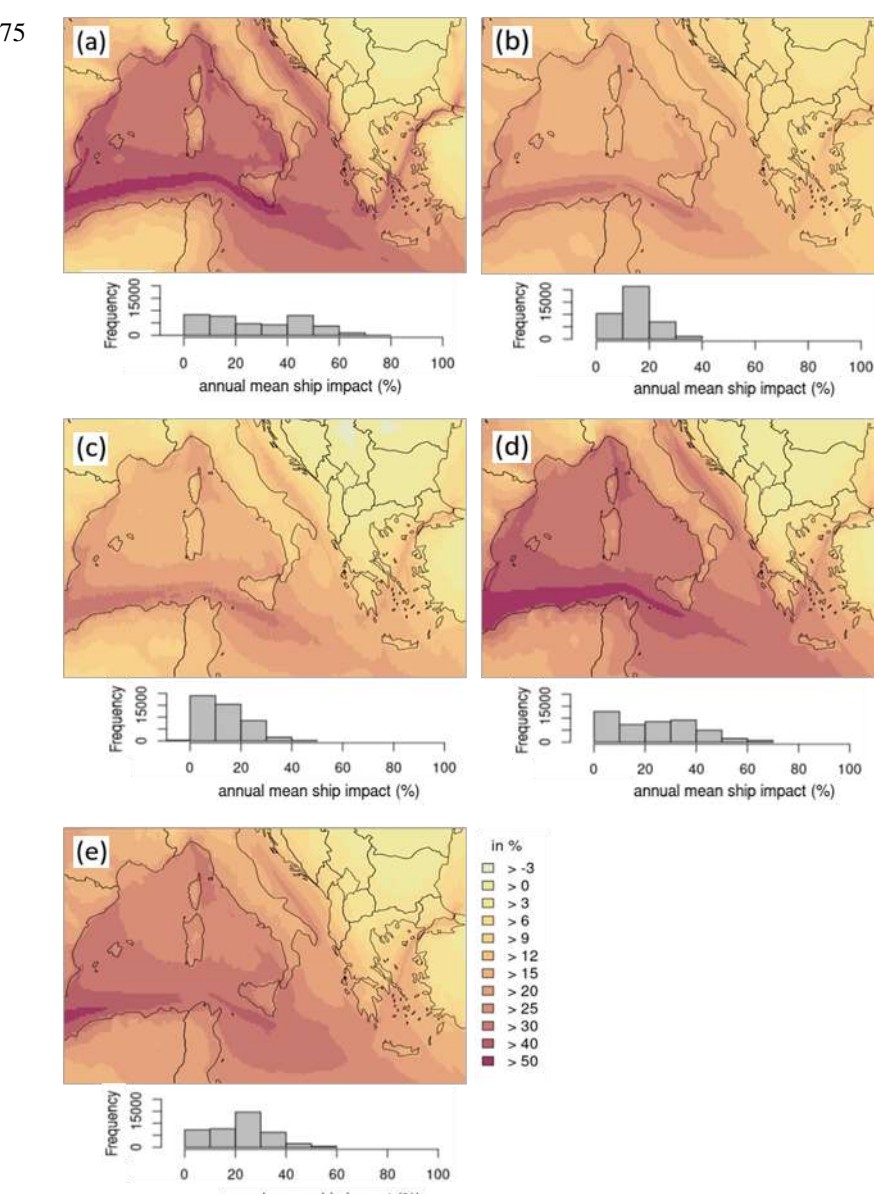

**Figure 6: Annual mean SO$_4^{2-}$ relative potential ship impact. (a) = CAMx, (b) = CHIMERE, (c) = CMAQ, (d) = EMEP, (e) = LOTOS-EUROS. Below the domain figure is the respective frequency distribution displayed for the annual mean SO$_4^{2-}$ potential ship impact, referred to the whole model domain.**





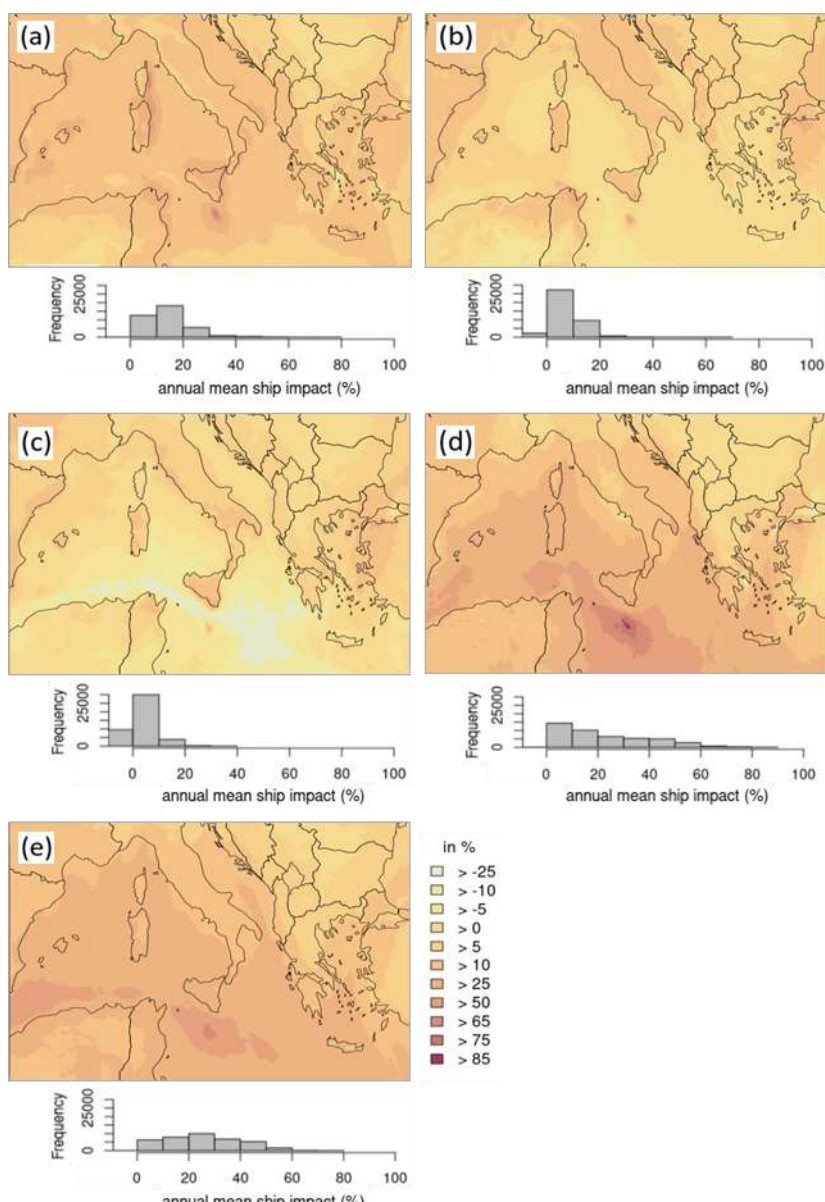

**Figure 7: Annual mean NO₃ relative potential ship impact. (a) = CAMx, (b) = CHIMERE, (c) = CMAQ, (d) = EMEP, (e) = LOTOS-EUROS. Below the domain figure is the respective frequency distribution displayed for the annual mean NO₃ potential ship impact, referred to the whole model domain.**





**Figure 8: (a) Boxplots for concentrations of PM$_{2.5}$, and the PM2.5 components SO$_4^{2-}$, NO$_3$, NH$_4^+$ and "others" as simulated by the five CTMs. The ensemble mean is "all_mean". Others is calculated as PM$_{2.5}$ minus the sum of SO$_4^{2-}$, NO$_3$ and NH$_4^+$. Data is based on the whole domain (all grid cells) and hourly data for all emission sources ("emisbase"). (b) Same as (a) but for ships only.**





**Table 4: Relative particle species of total PM$_{2.5}$ emissions.**

|  | Ensemble mean | CAMx | CHIMERE | CMAQ | EMEP | LOTOS-EUROS |
|---|---|---|---|---|---|---|
| **SO$_4^{2-}$** | 22.8 | 14.6 | 27.0 | 23.8 | 22.5 | 24.8 |
| **NO$_3$** | 8.0 | 11.1 | 3.1 | 14.5 | 5.6 | 10.6 |
| **NH$_4^+$** | 7.1 | 6.5 | 5.6 | 6.2 | 7.8 | 12.2 |
| **Other** | 62.1 | 67.8 | 64.3 | 55.5 | 64.1 | 52.4 |

**Table 5: Relative particle species of total shipping-related PM$_{2.5}$.**

|  | Ensemble mean | CAMx | CHIMERE | CMAQ | EMEP | LOTOS-EUROS |
|---|---|---|---|---|---|---|
| **SO$_4^{2-}$** | 44.6 | 37.0 | 36.0 | 48.5 | 63.9 | 51.8 |
| **NO$_3$** | 8.6 | 13.1 | 2.5 | 11.9 | 6.6 | 16.9 |
| **NH$_4^+$** | 12.4 | 11.7 | 9.1 | 12.6 | 11.8 | 23.5 |
| **Other** | 24.4 | 38.2 | 52.4 | 27.0 | 17.7 | 7.8 |



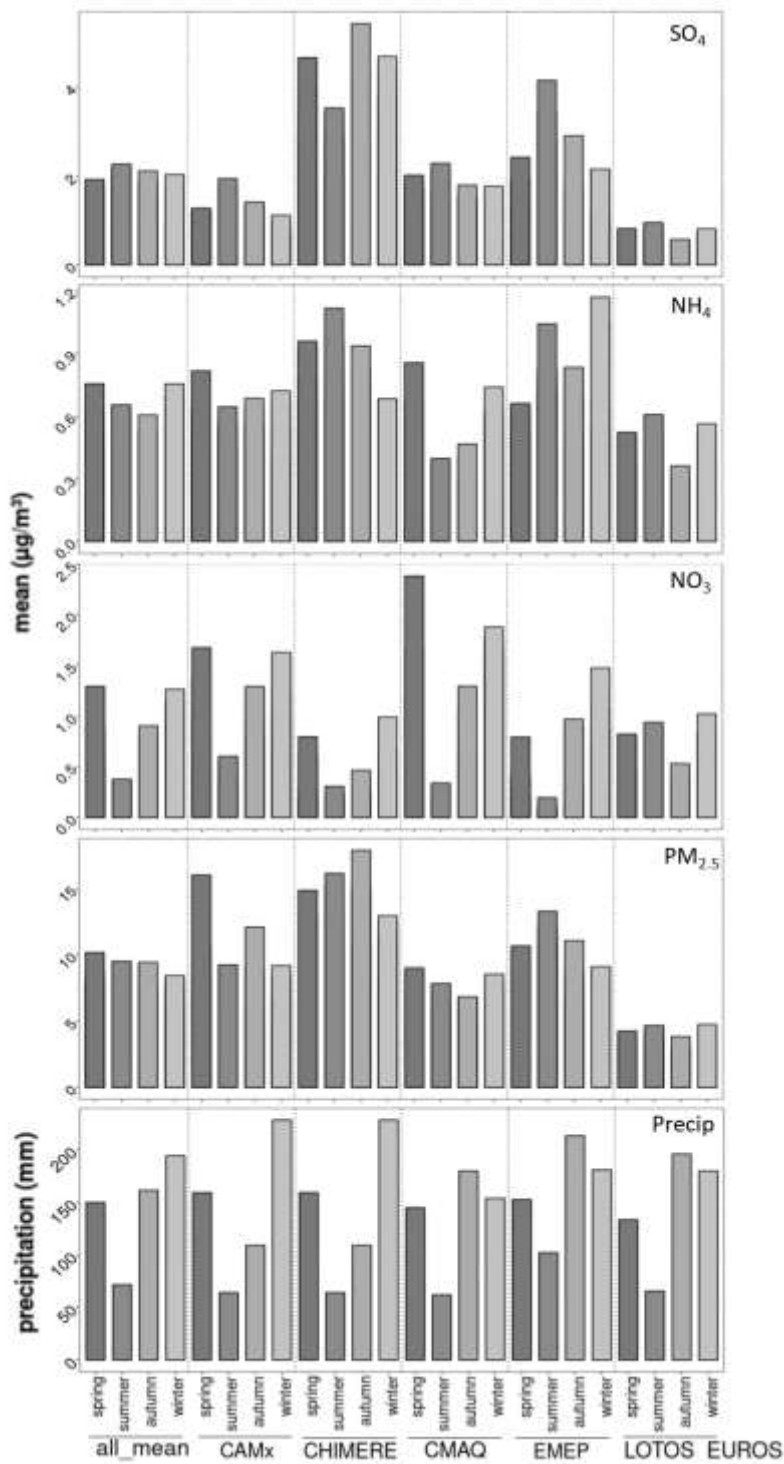

**Figure 9: Concentration of particle species and precipitation divided by seasons and CTMs. "all_mean" displays the model ensemble. Spring = March, April, May; summer = June, July, August; autumn = September, October, November; winter = December, January, February. Concentration is based on the annual median over the whole domain. Precipitation displays the seasonal sum (in mm).**





### 3.4.2 Wet Deposition

Wet deposition can provide hints about the fate of particles. EMEP does not deliver separate deposition files for individual particle species but for reduced and oxidized nitrogen. Thus, EMEP is not considered when analyzing wet deposition in this study.

Regarding spatial distribution of $NH_4^+$ wet deposition, highest annual sums are displayed by CMAQ and LOTOS-EUROS (up to 250 mg/m²/year over land; up to 50 mg/m²/year over water; Figure S21). CAMx and CHIMERE show a similar spatial distribution with values mainly between 10 mg/m²/year and 25 mg/m²/year. CAMx and CHIMERE used the same meteorology data, but despite of this the seasonal distribution of wet deposition differs (Figure 10).

Regarding the wet deposition of sulfate, the annual totals for all emission sources are highest over the Balkan Peninsula in the CMAQ and LOTOS-EUROS model outputs (300 mg/m²/year to 800 mg/m²/year; Figure S22). For CAMx over land areas, the values reach 300 mg/m²/year, and the lowest totals over land can be seen in the CHIMERE results (0.0 mg/m²/year to 50 mg/m²/year). Over water, these values are low in all model outputs (50 mg/m²/year to 150 mg/m²/year), except CHIMERE, contrary to the other models, highest wet deposition was found over water.

The wet deposition of $NO_3^-$ is highest for CMAQ (> 400 mg/m²/year) over the whole domain (Figure S23). For CAMx and LOTOS-EUROS, it is generally lower, with most areas displaying 25 mg/m²/year to 50 mg/m²/year. Lowest wet deposition of nitrate is shown in CHIMERE outputs with values not exceeding 50 mg/m². Regarding the sum for the whole year, the highest values are found for CMAQ (Northern Italy and the Balkan Peninsula, where the urban-area values reached 400 mg/m²/year). Over water, deposition is lower than over land in the results of all CTMs. Lower wintertime precipitation in CMAQ compared to the other models might lead to high particle concentrations as well as high deposition due to low dilution (Figure 10).

Wet deposition depends mainly on the ability of models to predict the amount, duration, and type of precipitation. The precipitation data show that the lowest values are found for CMAQ input data. CAMx and CHIMERE use the same meteorological input data and thus display the same precipitation results, with the highest values in winter. CMAQ and LOTOS-EUROS have precipitation values within a similar range, with the highest values occurring in autumn and winter.

Although the precipitation results in CAMx and CHIMERE are the same, wet deposition differed among these two models, indicating that the concentration as well as model internal mechanisms caused differences rather than the input data. Additionally, in CMAQ, a lower wet deposition rate is expected for nitrate. There are usually two mechanisms important for scavenging in CMAQ; in-cloud and below-cloud scavenging. High wet deposition for nitrate in CMAQ outputs might be traced back to efficient below cloud scavenging of coarse mode particles containing nitrate, through which the wet deposition can be high despite precipitation in similar ranges as other models. Furthermore, the deposition of particulate nitrate crucially depends on the reactive uptake of $HNO_3$ to larger particles (Karl et al., 2019), because coarse-mode particles are removed much faster than fine-mode particles.





**Figure 10: Wet deposition sum (mg/season) of particle species and precipitation divided by seasons and CTMs. "all_mean" displays the model ensemble. Spring = March, April, May; summer = June, July, August; autumn = September, October, November; winter = December, January, February. Wet deposition is based on the annual sum over the whole domain. Precipitation displays the seasonal sum (in mm).**



## 4 Discussion

Various reasons for underestimations in $PM_{2.5}$ in regional CTM systems might be traced back to model-specific calculations. For CAMx, Pepe et al. (2019) linked these underestimations to meteorological parameters and to the overestimation of the vertical mixing in the lower atmosphere. Tuccella et al. (2019) found underestimations of $PM_{2.5}$ in the CHIMERE model and explained these by an excess of wet scavenging in the model. An excess of wet scavenging in CHIMERE compared to the other CTM systems is not found in the present study, thus it cannot be used as explanation for deviations here. In EMEP, differing from the other CTM systems, the MARS module was used to calculate the equilibrium between the gas and aerosol phases; this model does not treat sea salt or dust, leading to underestimations of $PM_{2.5}$. Kranenburg et al. (2013) linked the underestimation of particulate matter in LOTOS-EUROS to the missing descriptions of SOA processes in the model. Thus, various reasons and combinations of reasons can lead to underestimations of $PM_{2.5}$ in the CTM systems used herein. For a better understanding, the inorganic particle species are considered in the present study. Consideration of inorganic as well as organic particles could lead to more uncertainties. Besides, in shipping emissions the inorganic aerosols display a higher share. Large part of $PM_{2.5}$ is secondary, therefore underestimations can be linked to underestimations of precursors, e.g., $NO_2$. This was already shown in the first part of this intercomparison study, where all five CTM systems underestimated measured $NO_2$ (Fink et al., 2023). But also $SO_2$ is usually underestimated by CTMs, as shown in previous studies (e.g. Eyring et al., 2007). Four out of five CTM systems underestimate the actual measured $PM_{2.5}$ concentration in the present study. Gaseous precursors like $SO_2$ and $NO_2$ need to be oxidized before they can form particles in reactions with ammonia. The hydroxyl radical (OH) is the main oxidant. The amount of available OH can be analyzed when the $NO_2$ concentration is set in relation to $HNO_3$ and $NO_3^-$ (Figure S24).This gives an indication about the OH availability. In ship plumes OH is consumed fast, therefore values are low along the shipping lanes. In regions with lower $NO_2$ concentration more OH is available and $HNO_3$ is efficiently formed. In the present study, the $HNO_3$ was similar within all five CTM systems (Figure S2).

One reason for the differences in $HNO_3$ might be traced back to the amount of cloud droplets, since $HNO_3$ is resolved in it. The dissolution of gases in droplets is usually assumed to be irreversible for $HNO_3$ and $NH_3$ in CTMs; thus, the amount of formed ammonium nitrate mass depends on the amount of $HNO_3$ or the cloud droplets. This could lead in the end to the deviation among the CTMs simulated $HNO_3$.

The preference of $NH_4^+$ to bind to $SO_4^{2-}$ in atmospheric aerosols to form $(NH_4)_2SO_4$ explains why in some models $NO_3^-$ displays relatively low values when the $SO_4^{2-}$ concentration is high. CHIMERE, for instance, has a $NO_3^-$ share of 3.1 % to total $PM_{2.5}$ and a $SO_4^{2-}$ share of 27.0 %, whereas in the CAMx results, $NO_3^-$ had a share of 11.1 % to total $PM_{2.5}$ and a $SO_4^{2-}$ share of 14.6 %. This can be confirmed by the low $SO_2$ concentration and high $SO_4^{2-}$ concentration in CHIMERE (Figures S9 & S17), indicating that sulfate is formed more efficiently compared to CAMx. Furthermore, this leads to lower $NO_3^-$ concentration in CHIMERE output (Figure S19). Also for $SO_2$ and $SO_4^{2-}$ concentration cloud water and amount of cloud droplets plays an important role.



Regarding the thermodynamic equilibrium within the models, ISORROPIA and ISORROPIA II mechanisms are used in all
CTM systems except EMEP, meaning similar results can be assumed to be obtained from this mechanism. Despite this
similarity, differences in concentrations may be a result of differences in available cloud water, vertical mixing, the
spatiotemporal distribution of emissions or aerosol size distributions. EMEP uses the MARS module to calculate the
equilibrium between the gas and the aerosol phase. Although four of five models use the ISORROPIA or ISORROPIA II
mechanisms for inorganic secondary aerosol formation, many factors within these models still cause significant differences
among the model outputs.

The aerosol size distribution also has an impact on the particle species distribution. As displayed in Table 1 (Section 2.1), there
are two concepts how the aerosol size distribution is represented within the models, either the distribution in bins or in log-
normal modes. As already discussed in Solazzo et al. (2012) the PM chemical composition differs greatly with the particle
size. Consequently, differences in modelling the aerosol size distribution also affects the chemical composition. In CMAQ, for
example, large fractions of nitrate and ammonium can be found in the coarse mode where they undergo other removal processes
than in the fine mode.

Although there is harmonization in terms of the input emission data in the present study, the internal model mechanisms used
to calculate particulate matter lead to differences in the particle species distribution, as discussed in Sect. 3.1.
In addition, the calculations how to determine $PM_{2.5}$ vary among CTM systems or even within one CTM. As an example, there
are two possibilities for calculating $PM_{2.5}$ within CMAQ: either online during the model run with the $PM_{2.5}$ module or
subsequently by calculating the value as the sum of two modes. These different options lead to different results (as shown by
Jiang et al., 2006) and will also affect the particle composition. In the present study, the sum of two modes is used in CMAQ.
Model simulations with relatively high $PM_{2.5}$ concentrations display higher absolute shipping impacts on $PM_{2.5}$, as presented
in Sect. 3.2. Consequently, relatively low variability in the relative potential ship impacts among the models compared to that
of the absolute values could be expected. For a more quantitative evaluation, relative potential ship impact is plotted against
absolute potential impact. A larger incline of the regression line can be explained by a higher background $PM_{2.5}$ concentration,
thus relative ship impact is lower for the same concentration increase (e.g. EMEP and CHIMERE) (Figure S25).

From the ISORROPIA and ISORROPIA II mechanism, it can be expected that the molar ratios between the acids on the one
side ($NO_3^-$ and $SO_4^{2-}$) and the base on the other side ($NH_4^+$) are in balance. However, the ratio between $SO_4^{2-}$, $NH_4^+$ and $NO_3^-$
shows that the balance in all models expect LOTOS-EUROS is not given for $PM_{2.5}$; sulfate plus nitrate is much higher
compared to ammonium (Figure S26). This balance is almost perfectly given in LOTOS-EUROS, although both, CMAQ and
LOTOS-EUROS, used the ISORROPIA II mechanism. Especially at the shipping lanes, an imbalance among the inorganic
particle species is present. Differences of particle species ratio among the models can be traced back to the differences in
particle size distribution. Contrary to the other models, CAMx has only three species in the coarse mode: coarse others primary,
coarse crustal and reactive gaseous mercury. For $NO_3^-$ and $SO_4^{2-}$, the ratio between the fine and coarse mode is calculated for
the CTMs (Figure S27 & S28). $NH_4^+$ was not considered here, since it is only in present in coarse mode in CMAQ.
These ratios show that CHIMERE and LOTOS-EUROS have only a small proportion of particles in coarse mode. For $SO_4^{2-}$





in LOTOS-EUROS the coarse particle concentration is zero and for EMEP no $SO_4^{2-}$ is present in coarse mode. In CMAQ a higher concentration of particles is assigned to the coarse mode, also for $NH_4^+$.

Regarding PM coarse and fine, another uncertainty among models might be caused by the differences in calculation of sea salt and dust emissions. Here again, both is considered in all CTMs, expect for dust in CMAQ. If sodium chloride and dust components are not considered, underestimations of PM and uncertainties in areas near coasts (sea salt) or where dust is important, e.g. Saharan dust in the Mediterranean region, occur, as described in Section 3.1. Furthermore, if sea salt and dust are omitted from the pH calculations, it might also cause deviations in sulfur chemistry, as this factor is very sensitive to pH.

In the CMAQ runs dust was considered at the model boundaries but dust emissions were not included, which limits the accuracy of predictions of both the total PM since the Mediterranean region is frequently affected by Saharan desert dust (Palacios-Peña, 2019).

The present study has shown that different reasons can cause deviations among the simulated $PM_{2.5}$ CTM outputs. Major reasons are the differences in size distribution and how models distribute chemical species among the coarse and fine mode

($PM_{2.5}$ and $PM_{10}$). Differences among the modelled $PM_{2.5}$ concentrations can also be a result of the differences in the height of the lowest model layer and the way in which ship emissions are distributed among the layers. As shown in Fink et al. (2023) the vertical distribution of $PM_{2.5}$ precursor emissions varies among the models, e.g. in CAMx all shipping emissions are assigned to the lowest layer. This leads to differences in chemical transformations because of different concentration levels close to the source and consequently to deviations among the particle distributions. Furthermore, precipitation differences lead

to variations among the model outputs for wet deposition.

Limitations of the present study are that only the chemistry of the lowest layer is evaluated. The model input was standardized as far as possible, but meteorological input data varied and is not compared in detail here. Interactions between fine and coarse particles are only studied to a limited extent, the same holds for aqueous chemistry, which has an impact on oxidation mechanisms of sulphur species.




## 5 Summary and Conclusion

The current work investigates and analyzes the predictions of five different CTM systems for $PM_{2.5}$ and inorganic particle species ($NH_4^+$, $SO_4^{2-}$, $NO_3^-$) dispersion and transformation in the Mediterranean region. Beside the total concentration focus is laid on the potential ship impact. Results show that four of the five models underestimated the actual measured $PM_{2.5}$
concentrations at stations close to the European coastline. The relative ship impacts on $PM_{2.5}$ simulated by the CTMs at the measurement stations are between 5.7 % (CMAQ) and 13.8 % (CAMx). The potential impact of $PM_{2.5}$ from ships simulated by CAMx, LOTOS-EUROS and EMEP have the largest areas with values up to 25 % along main shipping routes in the Mediterranean Sea. CMAQ and CHIMERE simulated potential ship impacts of 15 % along the main shipping lines close to the African coast. These impacts are within the range of the ship impacts obtained in other studies.

The spatial distribution of ammonium displays a low total annual mean mainly in the southwestern part of the domain (approximately 0.0 µg/m³) and is highest in the Po Valley and Bosporus (1.5 µg/m³). The ensemble mean of $NH_4^+$ (0.6 µg/m³) is in good agreement with the measurements provided in previous studies. The relative and absolute ship impacts are very similar for all models (0.0 µg/m³ to 0.06 µg/m³ over land, up to 0.15 µg/m³ over water; 0.25 % to 5.0 % over land, and 10 % to 25 % over water). This indicates that differences among the simulated $PM_{2.5}$ from ships result from differences in sulfate
and nitrate.

The $NH_4^+$ proportion to total $PM_{2.5}$ is similar in all models (5.6 % to 7.8 %), and only LOTOS-EUROS shows a relatively high share (12.2 %). The ship impact pattern is similar; all models display proportions between 9.1 % and 12.6 %, but higher values are simulated by LOTOS-EUROS (23.5 %).

$SO_4^{2-}$ is main contributor to the total $PM_{2.5}$ concentration regarding shipping emissions only. In the model ensemble mean for
the absolute ship concentration, $SO_4$ 2- accounts for 44.6 % of $PM_{2.5}$. The annual mean sulfate total concentration is highest for CHIMERE in the eastern part of the domain, reaching 6.0 µg/m³. CAMx, CMAQ, EMEP and LOTOS-EUROS simulate a total $SO_4^{2-}$ concentration within one range between 0.4 µg/m³ and 2.0 µg/m³ in the western part of the domain. The relative potential ship impacts on total $SO_4^{2-}$ are lowest over land, with values up to 3.0 %, and higher in coastal areas, with values ranging from 6 % to 20 %. Along the main shipping routes, the impacts are highest, reaching 50 % for CAMx, EMEP and
LOTOS-EUROS; for CHIMERE and CMAQ, they are lower, with values reaching 30 %. Regarding the absolute ship impacts on $SO_4^{2-}$, the model simulations display similar concentrations and are slightly lower for CMAQ and LOTOS-EUROS. Especially over water areas with relatively high $SO_2$ and $SO_4^{2-}$ concentrations are identified. Because $NH_4^+$ preferentially binds to $SO_4$ 2- in atmospheric aerosols to form $(NH_4)_2SO_4$, in areas over water less $NH_4NO_3$ forms.

The highest annual mean $NO_3^-$ total concentrations appear over land areas in the simulations by CAMx, CMAQ and LOTOS-
EUROS, especially over Italy and in the Balkan states (> 2 µg/m³). The lowest concentrations are simulated by CHIMERE. The concentrations over water are lower than those over land areas. The ensemble mean of all CTMs over the whole domain shows a median value of 0.63 µg/m³. Higher concentrations over land than over water are expected due to $NH_4NO_3$ formation in areas characterized by high $NH_3$ and $HNO_3$ conditions and low $SO_4^{2-}$ conditions.



The relative potential ship impact on $NO_3^-$ differs among the models. The CAMx, EMEP and LOTOS-EUROS results are similar; the relative potential ship impacts over land ranges from 0.0 % to 5.0 % (in the Balkan states), those in coastal areas and Italy ranges from 10 % to 25 % and those along main shipping routes from 50 % to 65 % or reaches even 85 %. CHIMERE and CMAQ show lower relative potential ship impacts for $NO_3^-$. For CMAQ, the impacts are lowest along the main shipping routes, nitrate is even reduced by 25 %. Low values in nitrate can be explained by the preference to form $(NH_4)_2SO_4$, thus nitrate stays in the gas phase or is transferred to the coarse mode. These low values for $SO_4^{2-}$ und $NO_3^-$ in CMAQ but relatively high values for EMEP, CAMx and LOTOS are reflected in the $PM_{2.5}$ ship impacts and partly explain the deviations in $PM_{2.5}$ among the models. As expected the seasonal variabilities in particle species show that $SO_4^{2-}$ and $NH_4^+$ are less temperature-dependent than $NO_3^-$. Nitrate is higher in winter and spring, but lower in summer and autumn. This pattern is found in all CTM simulations.

The spatial distribution of $NH_4^+$ wet deposition shows highest annual sums by CMAQ and LOTOS-EUROS (up to 250 mg/m²/year over land; up to 50 mg/m²/year over water). CAMx and CHIMERE show a similar spatial distribution with values mainly between 10 mg/m²/year and 25 mg/m²/year. For wet deposition of $SO_4^{2-}$, the annual totals for all emission sources are highest over the Balkan Peninsula in the CMAQ and LOTOS-EUROS model outputs (300 mg/m²/year to 800 mg/m²/year). For CAMx over land areas, the values reach 300 mg/m²/year, and the lowest totals over land can be seen in the CHIMERE results (0.0 mg/m²/year to 50 mg/m²/year). Over water, these values are low in all model outputs (50 mg/m²/year to 150 mg/m²/year), except CHIMERE. The wet deposition of $NO_3^-$ is highest for CMAQ (> 400 mg/m²/year) over the whole domain. For CAMx and LOTOS-EUROS, it is generally lower, with most areas displaying 25 mg/m²/year to 50 mg/m²/year. Lowest wet deposition of nitrate is shown in CHIMERE outputs with values not exceeding 50 mg/m². Over water, deposition is lower than over land in the results of all CTMs.

The complexity of particle treatments within the models, as well as the large number of causes of these changes make it difficult to find a single cause for the variable outputs. One point causing uncertainties is that the aerosol-formation mechanisms differ among CTMs. The detailed investigation of $PM_{2.5}$ and its chemical composition has demonstrated that differences among the particle species might be traced back to the aerosol size distribution. This was shown especially for CMAQ regarding the balance of the inorganic particle species nitrate and sulphate on the one side and ammonium on the other side. CMAQ and EMEP tend to assign a higher particle mass to the coarse mode compared to the other three CTMs. This has implications for particle deposition because both, wet and dry deposition are more efficient for larger particles.

An ensemble mean with standard deviations based on several model results can provide a more reliable assessment of possible ship impacts on air concentration and deposition. Previous research has demonstrated that using only one chemical transport model resulted in underestimated model uncertainty and overconfidence in the conclusions (e.g., Solazzo et al., 2013; Riccio et al., 2012; Solazzo et al., 2018), indicating that a model ensemble should better be used. Particularly in terms of the study's policy point of view, the ensemble mean is important: If model simulations are used to support in decision-making regarding shipping regulations, the uncertainty of individual models must be considered.





Goal of this study was not to make model outputs as similar as possible, but to show the discrepancies that occur among CTM systems despite using similar input data. Different CTM systems were asked the same question to find the impact of shipping for which they got the same emissions as input data.

Nevertheless, to achieve less varying results in future studies, the vertical emission distribution as well as the boundary conditions could be the same in all CTM inputs. This can help to make the modeled output more alike. Adjustments in using the same meteorology could be also helpful yet difficult to realize, since the meteorology and meteorological driver within each CTM system is closely connected. To better insight in certain mechanisms, one model could be used with e.g. changing vertical profiles, emissions or meteorology.

Regional-scale models with relatively coarse grid resolutions do not account for chemical transformation mechanisms within a ship's exhaust gas plume. They typically assume direct dilution and neglect the in-plume chemistry at high pollutant concentration levels. To obtain more precise information regarding effects of shipping on particle concentrations, the particle size distribution and the interaction mechanisms from plume to background concentrations, as well as chemical transformations within ship plumes should be considered in future studies.



**Code and data availability**

CAMx source code and documentation can be downloaded from https://camx-wp.azurewebsites.net/download/source/ (last access: 19 January 2023; Ramboll, 2023) and the Chimere website (https://www.lmd.polytechnique.fr/chimere/2020_getcode.php, last access: 19 January 2023). CMAQ version 5.2, which was used here, is available at https://doi.org/10.5281/zenodo.1167892 (US EPA Office of Research and Development, 2017). EMEP is available at https://doi.org/10.5281/zenodo.3647990 (EMEP MSC-W, 2020), LOTOS-EUROS is available at https://lotos-euros.tno.nl/open-source-version/ (last access: 19 January 2023; TNO, 2023), and WPS/WRF is available from WPS (2022; https://github.com/wrf-model/WPS, last access: 19 January 2023) and WRF Community (2000, https://doi.org/10.5065/D6MK6B4K). The COSMO software is available at https://www.cosmo-model.org/content/support/software/default.htm#models (last access: 24 January 2023; COSMO, 2023) and ecmwf-ifs/ifs-scripts at https://github.com/ecmwf-ifs (last access: 19 January 2023; ECMWF, 2023).

Data on measurement stations from EEA can be downloaded at https://discomap.eea.europa.eu/map/fme/AirQualityExport.htm (last access: 20 January 2023). CTM model results are available upon request.

**Author contributions**

LF: CMAQ model runs, evaluation and analysis of model results, preparation and writing of the paper. MK: Analysis of the results, revision of the text. VM: Supervision, analysis of the results, revision of the text. SO: CAMx and CHIMERE model runs, discussion of the results. RK and JK: LOTOS-EUROS model runs, land-based emissions data provision, discussion of the results. JM and SJ: EMEP model runs, discussion of the results. JPJ and EM: STEAM model runs, shipping emissions data provision, discussion of the results.

**Competing interests**

The contact author has declared that none of the authors has any competing interests.

**Acknowledgements**

This was work supported by SCIPPER project, which has received funding from the European Union's Horizon 2020 research and innovation programme under grant agreement Nr.814893.
AtmoSud acknowledges the continuous support of CAMx by RAMBOLL and CHIMERE by LMD.



The Community Multiscale Air Quality Modelling System (CMAQ) is developed and maintained by the USEPA. Its use is gratefully acknowledged. Ronny Petrik from Helmholtz-Zentrum Hereon (now at Marinekommando Deutsche Marine, Rostock) is acknowledged for providing meteorology and boundary conditions for CMAQ runs.

The computations for the regional modeling using the EMEP-model was enabled by resources provided by the Swedish National Infrastructure for Computing (SNIC), partially funded by the Swedish Research Council through grant agreement no. 2018-05973.

Support from Meteorological Synthesizing Center – West of EMEP at Norwegian Meteorological Institute, especially Peter Wind and David Simpson, in implementation of emissions and meteorological fields used in this paper into EMEP Open Source model is gratefully acknowledged.

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



**Appendix A:**

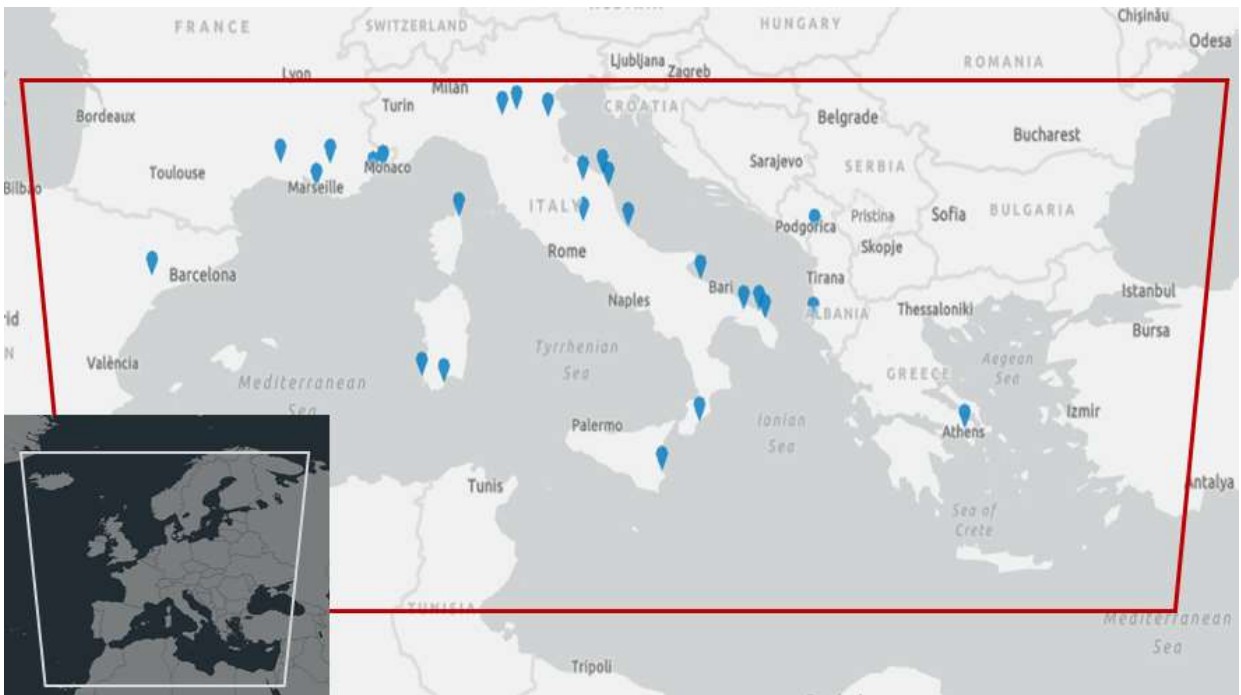

**Figure A1: Domains and measurement stations. Red trapeze displays the 12 x 12 km² domain, blue dots are locations of measurement stations. On bottom left the larger 36 x 36 km² domain is displayed.**



**Appendix B:**


**Table B1: detailed overview of monitoring stations**

| Name | Code | Country | Latitude | Longitude | Elevation | Station Type | Data Points | Measured Pollutants |
|---|---|---|---|---|---|---|---|---|
| **Vlora** | al0204a | Albania | 40.40309 | 19.4862 | 25 | urban background | 6850 | benzene, CO, $NO_2$, $NO_x$, $O_3$, $PM_{10}$, $PM_{2.5}$, $SO_2$ |
| **Shkoder** | al0206a | Albania | 42.3139 | 19.52342 | 13 | urban background | 7536 | CO, $NO_2$, $NO_x$, $O_3$, $PM_{10}$, $PM_{2.5}$, $SO_2$ |
| **Els Torms** | es0014r | Spain | 41.39389 | 0.73472 | 470 | rural background | 8549 | NO, $NO_2$, $NO_x$, $O_3$, $SO_2$, $PM_{2.5}$ |
| **Marseille 5 Avenues** | fr03043 | France | 43.30607 | 5.395794 | 73 | urban background | 8585 | $NO_2$, $O_3$, $PM_{10}$, $PM_{2.5}$, $SO_2$ |
| **Gauzy** | fr08614 | France | 43.8344 | 4.374219 | 40 | urban background | 8406 | $NO_2$, $O_3$, $PM_{10}$, $PM_{2.5}$ |
| **Cannes Broussilles** | fr24009 | France | 43.5625 | 7.007222 | 71 | urban background | 8587 | $NO_2$, $O_3$, $PM_{10}$, $PM_{2.5}$ |
| **Manosque** | fr24018 | France | 43.83527 | 5.785831 | 385 | urban background | 8517 | $NO_2$, $O_3$, $PM_{10}$, $PM_{2.5}$ |
| **Nice Arson** | fr24036 | France | 43.70207 | 7.286264 | 11 | urban background | 8701 | $NO_2$, $O_3$, $PM_{10}$, $PM_{2.5}$ |
| **Bastia Montesoro** | fr41017 | France | 42.67134 | 9.434644 | 47 | rural background | 8626 | $NO_2$, $O_3$, $PM_{2.5}$ |
| **Lykovrysi** | gr0035a | Greece | 38.06963 | 23.77689 | 210 | suburban background | 6719 | $NO_2$, $NO_2$, $PM_{2.5}$, $O_3$ |
| **Priolo** | it0614a | Italy | 37.15612 | 15.19087 | 35 | urban background | 7902 | $NO_2$, $PM_{2.5}$, benzene, $SO_2$ |
| **Leonessa** | it0989a | Italy | 42.5725 | 12.96194 | 948 | urban background | 8207 | $NO_2$, $PM_{2.5}$, $O_3$ |
| **Gherardi** | it1179a | Italy | 44.83972 | 11.96111 | -2 | rural background | 8269 | $NO_x$, $PM_{2.5}$, $NO_2$, $O_3$ |



| | | | | | | | |
|---|---|---|---|---|---|---|---|
| **Teatro d'Annunzio** | it1423a | Italy | 42.45639 | 14.23472 | 4 | urban background | 8135 | $NO_2$, $O_3$, $PM_{10}$, $PM_{2.5}$, $SO_2$, benzene, CO |
| **Cenps7** | it1576a | Italy | 39.20333 | 8.386111 | 25 | suburban background | 7968 | CO, $NO_2$, $SO_2$, $PM_{2.5}$ |
| **Lecce - S.M. Cerrate** | it1665a | Italy | 40.45889 | 18.11611 | 10 | rural background | 7290 | $NO_2$, $O_3$, $PM_{2.5}$ |
| **Brindisi Via Magellano** | it1702a | Italy | 40.65083 | 17.94361 | 10 | suburban background | 7904 | $NO_2$, $PM_{10}$, $PM_{2.5}$ |
| **Genga - Parco Gola della Rossa** | it1773a | Italy | 43.46806 | 12.95222 | 550 | rural background | 5310 | $NO_2$, $O_3$, $PM_{10}$, $PM_{2.5}$, $SO_2$, benzene, CO |
| **Civitanova Ippodromo S. Marone** | it1796a | Italy | 43.33556 | 13.67472 | 110 | rural background | 6699 | $NO_2$, $NO_x$, $O_3$, $PM_{10}$, $PM_{2.5}$, benzene |
| **Ancona Cittadella** | it1827a | Italy | 43.61167 | 13.50861 | 100 | urban background | 5985 | $NO_2$, $O_3$, $PM_{10}$, $PM_{2.5}$, benzene, CO, $SO_2$ |
| **Schivenoglia** | it1865a | Italy | 44.99694 | 11.07083 | 16 | rural background | 8325 | $NO_2$, $NO_x$, $O_3$, $SO_2$, benzene, $PM_{2.5}$ |
| **San Rocco** | it1914a | Italy | 44.87306 | 10.66389 | 22 | rural background | 8398 | $NO_2$, $NO_x$, $O_3$, $PM_{2.5}$ |
| **Locri** | it1940a | Italy | 38.22976 | 16.25518 | 11 | urban background | 8509 | $NO_2$, $O_3$, $SO_2$, benzene, CO, $PM_{2.5}$ |
| **Censa3** | it1947a | Italy | 39.06667 | 9.008889 | 56 | urban background | 8169 | $NO_2$, $SO_2$, benzene, $PM_{2.5}$ |
| **Stadio Casardi** | it2003a | Italy | 41.31667 | 16.28611 | 15 | urban background | 8391 | $NO_2$, $O_3$, benzene, $PM_{2.5}$ |
| **Ceglie Messapica** | it2148a | Italy | 40.64917 | 17.5125 | 100 | suburban background | 8393 | $NO_2$, $PM_{10}$, $PM_{2.5}$, $SO_2$, CO, benzene |