# Peer review of "A multimodel evaluation of the potential impact of shipping on particle species in the Mediterranean Sea"

_EGUsphere, 2023_

## Author Response (AR1)

**Answers RC 1**

The manuscript provides a relevant analysis of the overall shipping emissions impact on the Mediterranean basin and on the coastal areas that are characterized by significant population density. Moreover, the proposed analysis completes the previous paper focused on gas pollutants already published by the same group of authors on ACP (Fink et al., 2023).

The Authors specify that part of the set-up of the compared models is heterogeneous (including meteorology, boundary conditions and dust and sea salt treatment).

Nevertheless, in different part of the manuscript these features should be better discussed even because the presented material (e.g. sea salt concentration in Figure S1) clearly shows the impact of the modelling of PM component not directly tied to anthropogenic emissions.

Some of the different features as e.g. boundary conditions and biogenic emissions should be better described to let the reader understand if the used dataset are derived from 2015 larger scale model simulations, climatological datasets or observations.

Some comments discussing if and how the general underestimation of $PM_{2.5}$ concentration provided by the models can affect the evaluation of the shipping contribution would complete the proposed discussion.

It could be of general interest if the presented analysis of PM composition could be compared with data derived from measurements in small island potentially impacted by shipping emissions and long range transport (e.g. https://acp.copernicus.org/articles/19/11123/2019/).

➔ We added a paragraph on this: "Mallet et al. (2019) traced back higher $SO_4^{2-}$ in the eastern part of the domain due to westerly winds. In the present study, we found this higher concentration for $SO_4^{2-}$ in the eastern part of the Mediterranean as well. On Lampedusa, they found ammonium sulfate contributed 63 % to PM1 mass, followed by organics (Mallet et al., 2019)." (page 22, line 441-444)

**Detailed comments**

➔ Thank you for your helpful comments! We edited the manuscript and added further analysis considering wind speed and sea salt.

**Abstract**

**Page 2, line 39**

"…how models distribute among the coarse…" probably refer to the distribution of emissions.

➔ Changed to: "[...] and how models distribute **emissions** among the coarse and fine mode ($PM_{2.5}$ and $PM_{10}$)." (page 2, line 39)

**1 Introduction**

**Page 4, line 115**

The S before "At" should be probably cancelled.

➔ It is removed (page 4, line 115)

**2.1 Models**

**Page 5, line 139-141**

The sentence includes a repetition of the portion "used for all CTMs"

➔ Changed to: "The same shipping emissions data from STEAM (version 3.3.0.; Jalkanen et al., 2009; Jalkanen et al., 2012; Johansson et al., 2013; Johansson et al., 2017) were used for all CTMs." (page 5, line 139)

**Line 143**

A reference to Table 1 could be added here.

➔ Added after: "Land-based emissions (CAMS-REG, v2.0), grid projection (WGS84_lonlat), domain (Mediterranean Sea), grid resolution (0.1° × 0.1°, 12 × 12 km) and the modeled year (2015) were also consistent (**Table 1**)." (page 5, line 142)

**Table 1**

It should be clarified if BCs are derived from model results, climatologies or 2015 specific data. The description is rather clear for CMAQ and LOTOS, referring to CAMS products, not for the other models.

The description provided for EMEP BCs is not clear. Does the provided sentence mean the EMEP provides BCs based on observations, model results or both? Are those data specific to year 2015?

Concerning biogenic emissions, are MEGAN emissions calculated from 2015 meteorology or do they refer to different periods? Is it possible to provide a specification better than "calculated online" for EMEP and LOTOS?

The dust emissions description too should be improved.

➔ The description is changed as well as more detailed information on the BCONs, biogenic emissions and dust emissions are added to Table 1 (page 6, line 165)

**2.1.2 Wet Deposition Mechanisms**

**Page 9, line 215-217**

These sentences contain repetitions and can be merged.

➔ The following is removed: "In EMEP, wet scavenging is treated with simple scavenging ratios, taking into account in-cloud and sub-cloud processes." (page 9, line 215-216)

**2.2.2 Shipping Emissions**

**Page 10, line 246**

Do the Authors mean that both the mentioned lower levels are characterised by the same depth of 42m?

➔ Yes, both layers have a depth of 42 meters. We added: "The height of the lowest **and of the second** layer in CMAQ are 42 m for each." (page 10, line 246)

**2.3 Observational Data, Statistical Analysis and Analysis of Model Results**

**Page 11, line 266**

Fink et al., 2013 should probably be Fink et al., 2023

➔ Yes, this was a typo.

**3.1 PM2.5 Model Performance**

**Page 12, line 278**

It should be reminded that CMAQ has no dust contribution.

➔ We added a sentence considering this: "Contrary to the other CTM systems, CMAQ does not consider dust emissions, but dust coming from the boundary which can cause underestimations in $PM_{2.5}$." (page 12, line 279)

**3.2 PM2.5 Spatial Distribution**

**Page 15, line 317-318**

Can this behaviour be caused by the sea salt contribution? Is it tied to wind speed distribution?

➔ The previous figures of the spatial distribution of sea salt in Supplement 1 for EMEP and CMAQ did not include coarse sea salt particles. This was changed in the current version that shows $PM_{10}$.

➔ A paragraph is added considering this behavior (page 15, line 345-348); this paragraph also answers the questions on line 339-340 in the next paragraph but one

**Line 325-326**

Is it the particle chemistry that causes the largest differences or dust & sea salt have a major role as it could be guessed from the presented comparison?

➔ We didn't use the same dust and sea salt in all models, thus in the present study we cannot trace it back to only either particle chemistry or dust and sea salt emissions. This was not evaluated separately.
It can merely be attributed to the fact that different organic modules were used and different calculations for dust and sea salt. In order to be able to say exactly, one would have to carry out a study aiming at answering this question.
Since we now have different plots within a new figure in Supplements 1, it shows that the difference among the models is smaller than assumed previously.

**Line 339-340**

Did the Authors investigate the surface wind speed and its treatment impact? The sea salt scheme itself seems similar those implemented in the other models.

➔ Please note that the figures in Supplement 1 were not correct for EMEP and CMAQ! We changed with the correct figures for sea salt. The previous ones for EMEP and CMAQ did not

include coarse particles. This leads to smaller deviations among the maps of the spatial distribution.

➔ The paragraph is changed and information on wind speed and sea salt are added: "The sea salt concentrations partly gives an explanation for the differing $PM_{2.5}$ concentration distribution among the models. The annual mean sea salt (NaCl) concentration is highest in CHIMERE, which partly explains the high $PM_{2.5}$ absolute concentration (Figure S1). The LOTOS-EUROS sea salt displayed lowest concentrations, also the overall $PM_{2.5}$ concentration is lowest compared to the other CTMs. The sea salt concentration was highest (up to 7.0 µg/m³) over sea in areas with high mean surface wind speed for CHIMERE, CMAQ, EMEP and LOTOS-EUROS (Figure S2). This can be confirmed by the high correlation between wind speed and sea salt concentration that was evaluated at several points over water for CMAQ, EMEP and LOTOS-EUROS (Figure S3, Table S4). There was no indication that $PM_{2.5}$ concentration was lower in areas with higher wind speed. CAMx considers sea salt as fine PM; there are no coarse sea salt particles in CAMx." (page 15, line 345-348)

**3.4.2 Wet Deposition**
**Page 30, line 491-492**

What is the possible reason of this peculiar behaviour of CHIMERE only?

➔ We added: "The explanation for this differing behavior might be provided by the different scavenging mechanisms. In CHIMERE the in-cloud mechanism for deposition of particles is assumed to be proportional to the amount of water lost by precipitation. In CAMx, the in-cloud scavenging coefficient for aqueous aerosols is the same as for the scavenging of cloud droplets. Below the cloud, CHIMERE uses a polydisperse distribution following Henzig et al. (2006) whereas in CAMx for rain or graupel the collection efficiency is calculated as in Seinfeld & Pandis (1998). The other possible explanation is that all the emissions in CAMx are emitted in the lowest layer and in CHIMERE the emissions follow a vertical distribution depending on source." (page 30, line 502 to 508)

**4 Discussion**

**Page 32, line 518-527**

In should be reminded in the discussion that the analysis and comparison of PM mass results are affected by the relevant differences in dust treatment, sea salt modelling and from the used boundary conditions (including themselves dust and salt issues).

➔ A sentence is added: "The treatment of dust, sea salt and the used boundary conditions have an effect on the analysis and comparison of PM results, because these parameters part of the $PM_{2.5}$ formation differ among the models." (page 33, line 548)

**Page 34, line 585-592**

Which kind of boundary conditions are used in particular for dust and sea salt? i.e. model driven, climatological, etc.

This discussion should be moved at the beginning of the section because it affects the PM mass and not only the size distribution of particles.

➜ The paragraph was moved at the beginning of the discussion and information on the boundary conditions for dust and sea salt were added:

"Regarding PM (coarse and fine for sea salt), another uncertainty among models might be caused by the differences in calculation of sea salt and dust emissions. Here again, both is considered in all CTMs, expect for dust in CMAQ. If sodium chloride and dust components are not considered, underestimations of PM and uncertainties in areas near coasts (sea salt) or where dust is important, e.g. Saharan dust in the Mediterranean region, occur, as described in Section 3.1. Furthermore, if sea salt and dust are omitted from the pH calculations, it might also cause deviations in sulfur chemistry, as this factor is very sensitive to pH.

In the CMAQ runs dust was considered at the model boundaries but dust emissions were not included. The Mediterranean region is frequently affected by Saharan desert dust (Palacios-Peña, 2019), but the main source region for this dust emission is not included in the model domain, thus the dust coming from the boundary can be seen as sufficient for the CMAQ model run.

Generally, the boundary conditions for dust and sea salt in CAMx and CHIMERE were produced by offline models that are running on meteorological fields from GEOS-5, GEOS DAS and MERRA. For CMAQ and LOTOS-EUROS these boundary conditions were produced model driven within the boundary conditions calculations. Boundary conditions of EMEP are developed from climatological ozone-sonde datasets." (page 33, line 536 – 548)

**Page 37, line 675-679**

It could be considered for discussion the general suggestion to perform analysis of test cases with controlled and shared BCs, sea salt and dust emissions (e.g. externally provided), that could enable a more consistent investigation of model results.

➜ A paragraph is added: "Furthermore, the present study neither use the same boundary conditions, nor did the models use the same sea salt or dust emissions. For more consistent investigations of model results future intercomparison studies could be carried out with using the same boundary conditions, sea salt and dust emissions as input data." (page 39, line 725-728)

**Answers RC 2**

The paper "A multimodel evaluation of the potential impact of shipping on particle species in the Mediterranean Sea" analyses the contribution of shipping emissions to particulate air pollution in the Mediterranean Sea and hence also their impact on air quality in the coastal areas of this region. It contains interesting scientific outcomes from a recent international project. In particular, results of five different Chemistry Transport Models (CTMs) are presented and compared in detail. The statistical approach followed is reasonable, and the discussion allows highlighting performance characteristics of the five models that might be helpful for prospective future users. Therefore, this paper deserves been published (after revision), in spite of two weak points:

(1) The overall discussion concentrates almost exclusively on chemical aspects. Possibly this was unavoidable, given the fact that the five CTMs were used "in their standard setup", as the authors admit (line 143), the consequence being that both meteorological input and boundary conditions differed among the five model simulations. This complicates extremely the conclusive discussion of the results obtained, because – strictly speaking – the latter are not comparable.

(2) It is the firm opinion of the present reviewer that the PM2.5 levels emerging from shipping and all other activities in the study area are significantly influenced by the dynamics of the Atmospheric Boundary Layer (ABL). In several parts of the manuscript the authors touch upon the issue, but it is obviously difficult for them to judge to what extent a bias or a poor correlation are associated with the treatment of chemistry in the CTMs or with inappropriate assumptions with regard to ABL dynamics. In this context, one has also to question the representativeness of the measurements used for comparisons with model results.

Undoubtedly, to remedy completely the above points would mean for the authors to repeat the simulations (perhaps with a more uniform treatment of ABL dynamics in the five models) and then to write from scratch a totally different paper. This is not what the present reviewer suggests. Instead, the authors should revisit the analysis of their results and their discussion commenting on the possible influence of the different treatment of ABL dynamics (and aspects related to boundary conditions) in the five models, thus delivering a more comprehensive opinion on the causes of deviations detected. In this upgraded analysis and discussion, they should also address the representativeness issue of the measurements used for comparison purposes.

Concerning minor and more detailed remarks, and despite the fact that the paper is well written, there are several repetitions and typos, as well as few language errors that the authors will most probably wish to remediate in their revised paper version.

➔ Thank you for your valuable comments! Indeed the ABL dymanics and different treatments have an influence on the model outputs. Considering this issue, we expanded our analysis and looked at the ABL in detail. We added two figures to the Supplements (Figure S27 and S28) and the following paragraph to the manuscript:

"All models used offline meteorology in which the ABL heights were calculated. Annual medians of the atmospheric boundary layer heights at 4 PM and 4 AM were compared among the models. The comparison of spatial distribution of ABL heights at 4 PM and 4 AM shows that over water, the ABL heights have not much variability in all models (Figure S27 and S28). The lowest ABL height over water was used for CHIMERE. This corresponds to the high $PM_{2.5}$ concentrations simulated by this model over water. Over land, the comparison of spatial distribution at 4 PM to 4 AM display more variable ABL heights: during nighttime the ABL heights are up to 200 m whereas during daytime the heights increase to 1000 m or higher (Figure S27 and S28). Over land the input in CAMx, CHIMERE, CMAQ and LOTOS-EUROS has a higher median ABL at 4 PM whereas in EMEP it is contrary with showing highest median at 4 PM mainly over water areas. Yet, there was no large deviation in $PM_{2.5}$ concentration simulated by EMEP to concentrations received from other models. Generally, due to ABL dynamics deviations from measured to simulated data can be expected because measurement stations were chosen close to the coast, which leads to uncertainties. In these areas, the measurements are influenced by air masses either coming from water or coming from land. In addition, measured data was received from one measurement point, which is hardly representative for a whole grid cell of 12 x 12 km². " (page 33, line 549 to 561)

Concerning the mechanisms to receive the ABL heights, the stability of the atmosphere was derived from the bulk Richardson number in CHIMERE, EMEP and LOTOS-EUROS. CHIMERE derives the boundary layer height from several formulations based on atmospheric static stability. Under stable conditions, the height is approximated using Troen and Mahrt (1986). Under unstable conditions, it is approximated using a simplified Cheinet and Teixeira (2003) technique. In EMEP, the ABL height is calculated using a slightly modified bulk Richardson number, as described by Jerievi et al. (2010). The method is also quite close to Seibert et al. (2000)'s bulk Richardson number approach. Finally, the ABL height is smoothed in space using a second order Shapiro filter (Shapiro, 1970). The ABL height can not be less than 100 m or greater than 3000 m. In LOTOS-EUROS, the boundary layer height is defined as the highest level with an updraft vertical velocity greater than zero and a local Richardson number greater than 50. More turbulent mixing is required inside the cloud layer in a stratocumulus boundary layer, and the height is set equal to the convective cloud top level.